# A human model to deconvolve genotype-phenotype causations in lung squamous cell carcinoma

Julia Ogden[1], Robert Sellers[1], Sudhakar Sahoo[1], Anthony Oojageer[1], Anshuman Chaturvedi[2], Caroline Dive [1,3,4] & Carlos Lopez-Garcia [1,4] ✉

Tractable, patient-relevant models are needed to investigate cancer progression and heterogeneity. Here, we report an alternative in vitro model of lung squamous cell carcinoma (LUSC) using primary human bronchial epithelial cells (hBECs) from three healthy donors. The co-operation of ubiquitous alterations (*TP53* and *CDKN2A* loss) and components of commonly deregulated pathways including squamous differentiation (*SOX2*), PI3K signalling (*PTEN*) and the oxidative stress response (*KEAP1*) is investigated by generating hBECs harbouring cumulative alterations. Our analyses confirms that *SOX2*-overexpression initiates early preinvasive LUSC stages, and co-operation with the oxidative stress response and PI3K pathways to drive more aggressive phenotypes, with expansion of cells expressing LUSC biomarkers and invasive properties. This cooperation is consistent with the classical LUSC subtype. Importantly, we connect pathway dysregulation with gene expression changes associated with cell-intrinsic processes and immunomodulation. Our approach constitutes a powerful system to model LUSC and unravel genotype-phenotype causations of clinical relevance.

Lung cancers are a devastating group of neoplasms which broadly divide into two categories, small cell lung cancer (SCLC) and non-small cell lung cancer (NSCLC). NSCLC accounts for 85% of lung cancer cases and primarily consists of two histologically distinct subtypes including lung adenocarcinoma (LUAD) (50% of all lung cancers) and lung squamous cell carcinoma (LUSC) (30% of all lung cancers)[1]. LUSC is a complex disease which predominantly originates in the proximal airways. Basal cells, which function as the multipotent progenitors of the bronchial epithelium, are believed to be the most likely cells-of-origin[2], although a more complex interplay between genetic drivers and cell-of-origin has been observed[3,4].

Contrasting with LUAD, patients with LUSC are almost exclusively smokers, their survival is poorer, and targeted treatments have largely failed[5–7]. LUSC exhibits a heterogeneous landscape of phenotypic subtypes with unclear biological origins[4,8]. Genetically, LUSC is characterised by inactivation of *TP53* and upregulation of the CDK4/6 pathway (typically by *CDKN2A* inactivation) in most cases[8,9], and unlike LUAD, dysregulation of oncogenic pathways in LUSC is less clear[9]. LUSC genetic landscapes have revealed components of the squamous differentiation (SD) pathway, PI3K/Akt signalling, and oxidative stress response (OSR) pathway to be the most frequently targeted, although co-occurrence or mutual exclusivity patterns in the dysregulation of these common LUSC pathways were not observed[9]. More recent multi-platform characterisations of LUSC have shown characteristic genomic features in specific subtypes[8,10]. Modelling this complexity is required to understand the underpinning biology, the specific vulnerabilities associated with each subgroup and to identify essential components of LUSC development.

[1]Cancer Research UK Manchester Institute, Wilmslow Road, M20 4BX Manchester, United Kingdom. [2]Department of Histopathology, The Christie Hospital, Wilmslow Road, Manchester M20 4BX, United Kingdom. [3]Cancer Research UK, National Biomarker Centre, Wilmslow Road, M20 4BX Manchester, United Kingdom. [4]Cancer Research UK Lung Cancer Centre of Excellence, Wilmslow Road, M20 4BX Manchester, United Kingdom. ✉ e-mail: carlos.lopezgarcia@cruk.manchester.ac.uk

Modelling LUSC heterogeneity in vivo has the obvious advantage of encompassing the tumour microenvironment (TME), but it is arduous, time-consuming, and costly. Non-immortalised primary human bronchial epithelial cells (hBECs), are a suitable and infrequently explored alternative. They can be expanded in vitro as basal cells and recapitulate bronchial epithelial architecture using organotypic systems[11]. Regardless of the absence of an autochthonous TME, hBECs are an alternative to mouse models in the deconvolution of complex genotypes to find genotype-phenotype causation, identify the essential alterations driving the cell-autonomous biology of LUSC development, uncover epistatic interactions between somatic events, and downstream actionable pathways with therapeutic application.

In this report, we aim to establish the potential of genetically engineered hBECs from three healthy donors to model LUSC and deconvolve the cell-intrinsic mechanisms whereby somatic events fuel LUSC progression using hBECs from three donors. We confirm that the SD, PI3K/Akt and OSR pathways are all necessary for LUSC development and invasive progression, identify pro- and anti-oncogenic interactions between those pathways and matched somatic alterations with dysfunction of biologically and therapeutically actionable pathways.

## Results

### hBEC genetic-engineering strategy to model LUSC

To determine the functions and cooperation of the most frequently dysregulated pathways (Fig. 1a) in LUSC development, we designed a combinatorial strategy in which a regulator of each pathway (*SOX2*, *PTEN* and *KEAP1*) was targeted individually, or in combinations of two and three (Fig. 1a). As *TP53* mutations and CDK4/6 pathway activation (typically by *CDKN2A* loss)[8,9] are ubiquitous in LUSC, we included concomitant *TP53* and *CDKN2A* (TC) truncations (Fig. 1a). To capture m phenotypic differences caused by inter-person variation and avoid smoking-induced pre-existing alterations, mutants were generated using hBECs from three never-smoking donors without reported airways disease (Supplementary Table 1). Genetic manipulation was achieved by electroporation with multiplex CRISPR-Cas9 constructs, selecting *TP53*[-/-] cells with Nutlin-3A, and lentiviral transduction to elevate the expression of *SOX2* and mimic its amplification (Fig. 1b, Supplementary Table 2 and Supplementary Fig. 1a, b). This strategy results in mutant hBECs with up to five modifications (Fig. 1a). CRISPR efficiency analysis revealed 96-100% indel efficiency and almost complete abrogation of the protein (Fig. 1c–f). Lentiviral *SOX2* resulted in *SOX2* overexpression (Fig. 1g) and analysis of SD, PI3K/Akt and OSR pathway surrogates demonstrated the expected pathway activation (Fig. 1e, f, h). We tested nine *PTEN* sgRNAs (Supplementary Table 2), only one successfully targeted the locus and this sgRNA also targets *PTENP1* (Supplementary Fig. 1c), a pseudogene with 98% *PTEN* homology. *PTENP1* encodes a lncRNA with miRNA decoy functions[12]. However, as *PTENP1* indels are restricted to *PTEN* mutants, do not disrupt miRNA binding sequences, and do not change the levels of *PTENP1* RNA (Supplementary Fig. 1d), we did not anticipate any functional consequences. Analysis of the top 5 predicted off-target sites for the remaining other sgRNAs did not show any indels.

### *SOX2* inhibits proliferation in 2D cultures but enhances anchorage-independent growth and invasiveness in cooperation with the OSR and PI3K/Akt pathway

Firstly, we assessed the transforming potential of our different mutant combinations in the three donors by investigating changes in proliferation, anchorage-independent growth, and invasiveness. Overall, mutants without lentiviral *SOX2*[OE] proliferated 2-3 fold faster than wild type hBECs and *PTEN* truncations enhanced this phenotype (Fig. 2a). However, *SOX2*[OE] mutants grew with similar rates to wild type, excluding TC + PS in donors 1 and 2 (Fig. 2a). As *SOX2*[OE] was not associated with increased apoptosis (Supplementary Fig. 1e), this result was

likely due to reduced proliferation in culture. Although paradoxical, *SOX2*[OE] mediated growth suppression has been observed in other systems[13,14] and indicates that SOX2 drives oncogenesis by mechanisms unrelated to proliferation. Indeed, only mutants with *SOX2*[OE] showed detectable anchorage-independent growth over background with TC + PKS displaying maximal colony formation in soft agar (Fig. 2b, c), demonstrating that SOX2[OE] is required for anoikis override and synergises with *PTEN* and *KEAP1* truncations. However, in donor 3, colony formation in TC + KS was comparable to TC + PKS. When we carried out invasion assays using the collagen disc method[15], an assay that incorporates the presence of pulmonary fibroblasts[15], we found that TC + PKS mutants showed the most invasive behaviour in all the donors (Fig. 2d, e), consistent with our soft-agar results (Fig. 2b, c). These results align with published in vivo evidence that highlights the essential role of *SOX2* in LUSC and indicate an oncogenic role independent of proliferation. Furthermore, our results confirm the cooperation of *SOX2* with activation of the PI3K/Akt and OSR pathways. Overall, these data characterising the newly generated models reflect the carcinogenic transformation of hBECs and encouraged further analyses.

### Effect of pathway dysregulation in bronchial epithelial homoeostasis

LUSC developmental stages are characterised by epithelial changes that include loss of specialised bronchial cells, surface maturation (squamous cells) and expansion of p63-expressing cells (Fig. 3a). To ascertain the role of pathway dysregulation in the development of these epithelial perturbations, we used organotypic air-liquid interface (ALI) cultures (Fig. 3b), widely used to study bronchial epithelial biology. Histological analysis revealed characteristic epithelial alterations in ALI cultures (Fig. 3c–f and Supplementary Fig. 2a, b). In general, wild-type and TC mutants lacked significant differences in epithelial structure and relative abundance of specialised bronchial lineages (goblet, ciliated and club cells) (Fig. 3c–e and Supplementary Fig. 2a, b). TC + P mutants showed a prominent phenotype characterised by increased cellularity and development of intra-epithelial cysts (Fig. 3c and Supplementary Fig. 2a), although without relative changes in differentiated bronchial lineages (Fig. 3d, e). TC + K and TC + PK mutants exhibited an overall reduction in the fraction of club cells (Fig. 3e), which suggests a previously unreported function of the OSR pathway in the differentiation dynamics of that lineage.

*SOX2*[OE] mutants exhibited abrogation of specialised cell differentiation except for TC + PS, which retained higher levels of differentiation (Fig. 3d, e). *SOX2*[OE] mutants also displayed evidence of squamous differentiation as indicated by apical dyskeratosis, most prominent in TC + S (Fig. 3c and Supplementary Fig. 3a), and expression of the squamous differentiation markers *SPRR2A*, *SPRR3* and *IVL* (Fig. 3f and Supplementary Fig. 2b). TC + PKS mutants developed keratinisation and a highly disorganised histology with frequent loss of basal-apical polarity, indicative of progression to more advanced developmental stages (Fig. 3c and Supplementary Fig. 2a). We reasoned that lower *SOX2* expression might be responsible for the comparatively abnormal behaviour of TC + PS mutants. Indeed, we found a discontinuous pattern of mCherry expression in TC + PS mutants (Fig. 3g), indicating negative selection of *SOX2*[OE] cells in TC + PS that is prevented by *KEAP1* inactivation.

LUSC developmental stages are characterised by an expansion of p63-expressing cells from the basal compartment to occupy the epithelial layer. Therefore, we predicted an expansion of p63-positive cells that would be maximal in our modelling of the most advanced stages of LUSC. To test this, we first confirmed the expansion of p63[+] cells in patient specimens with low and high-grade premalignant lesions (Fig. 4a, b). Tissue regions with premalignant lesions were divided into basal and apical halves and the fraction of total p63[+] cells residing in the apical half was quantified as a measure of p63[+] cell

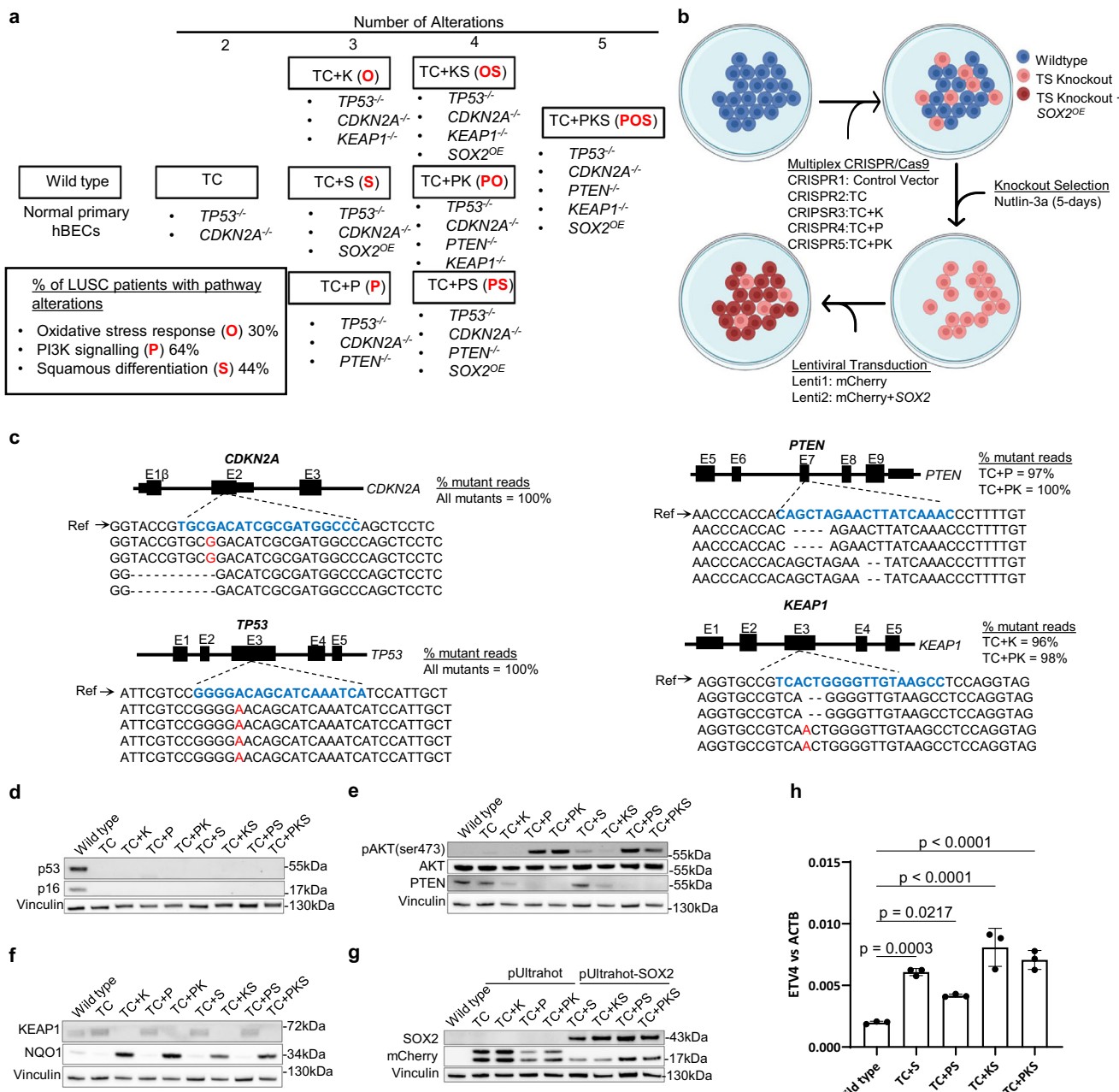

**Fig. 1 | Experimental design and genetic modification strategy of hBECs. a** The combinatorial approach to modelling LUSC pathway dysregulation in primary hBECs. Eight mutant groups with increasing numbers of alterations were generated. All mutants harboured disruption of the *TP53* and *CDKN2A* genes. *KEAP1*, *PTEN* and *SOX2* alterations were included to target the oxidative stress response, PI3K signalling and squamous differentiation pathways, respectively. T = *TP53*, C = *CDKN2A*, P = *PTEN*, K = *KEAP1*, S = *SOX2*. Red letters indicate the pathways targeted in each mutant. O = the oxidative stress response, P = PI3K signalling, and S = squamous differentiation. **b** Gene editing strategy for the disruption of tumour suppressor genes by electroporation with multiplex CRISPR/Cas9 vectors and lentiviral mediated *SOX2* overexpression. Five electroporations were carried out per donor: Control vector = CRISPR/cas9 vector lacking gRNA, TC = *TP53/CDKN2A* gRNAs, TC + P = *TP53/CDKN2A/PTEN* gRNAs, TC + K = *TP53/CDKN2A/KEAP1* gRNAs, TC + PK = *TP53/CDKN2A/PTEN/KEAP1* gRNAs. Those mutants with *SOX2*[OE] were transduced with the pUltrahot vector carrying an mCherry reporter and *SOX2* cDNA and those without *SOX2*[OE] were transduced with the empty pUltrahot vector with mCherry reporter. **c** Next generation amplicon sequencing of CRISPR/Cas9 target loci for each gene of interest. Example mutant reads are aligned to a wild type reference sequence. gRNA target sequences are identified in blue. Red text indicates nucleotide insertions. Deleted single nucleotides are indicated (-). Right panel displays % mutant reads. **d**–**g** Tumour suppressor knockouts, mCherry and SOX2 protein expression were confirmed by western blotting for target genes. pAKT(ser473) and NQO1 were used as surrogates for activation of PI3K signalling and the OSR, respectively. This analysis was also performed for the other two donors. Panels **d**, **e** and **g** contain loading controls for all targets. Vinculin in panel f is the loading control for NQO1 and sample processing control for KEAP1, which was run in the blot for panel d. Panels d and f contain the same loading control. Total AKT and pAKT (ser478) were re-probed after sequential stripping of the membrane. **h** qPCR analysis of the *SOX2* target *ETV4* in mutants with SOX2 overexpression. Mean +/- SD (*n* = 3 ALI cultures from donor 1). Adj.P values were calculated by one-way ANOVA with multiple comparisons and Dunnett's post *hoc test*. ns = not significant (95% CI). Figure 1b created in BioRender[52]. Source data are provided as a Source Data file.

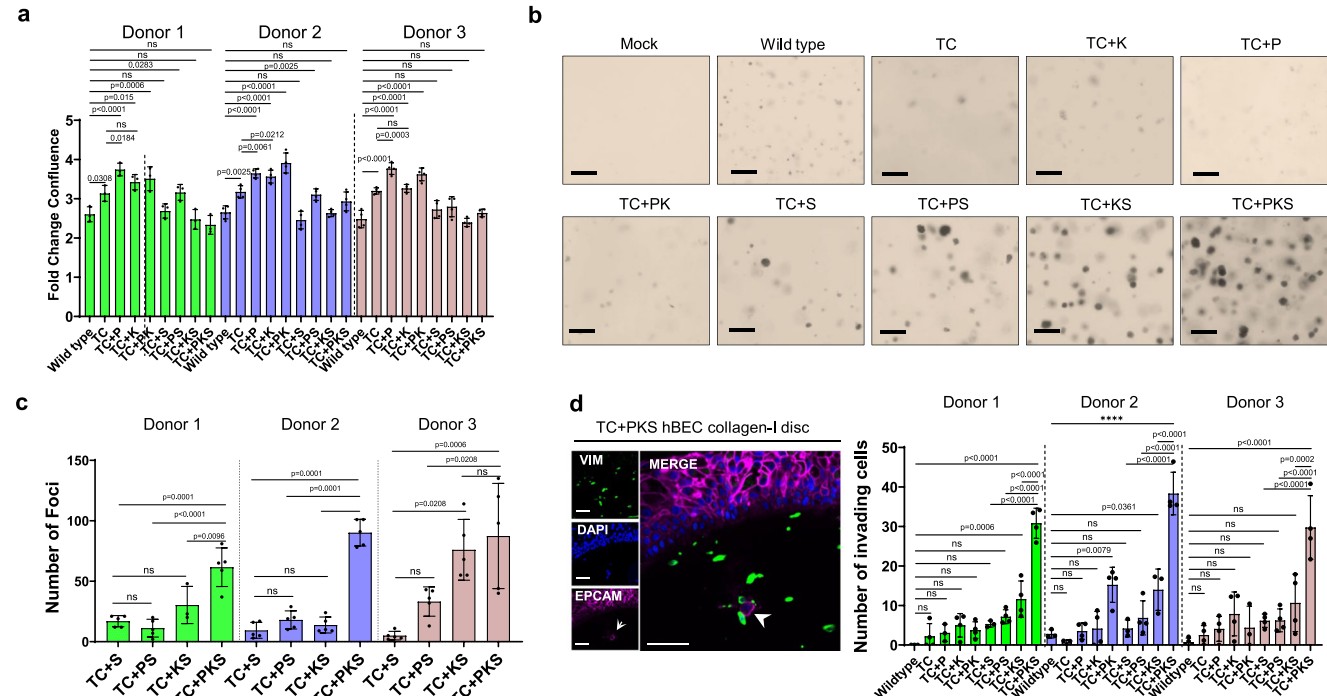

**Fig. 2 | Phenotypic analysis of mutant hBECs. a** Cell population growth assays for wild type and mutant hBECs in three donors. Data is shown as the fold-change in confluence from 24-72 h. Mean +/- SD (n = 4 independent biological replicates). Adj.P values were calculated by one-way ANOVA with multiple comparisons and Holm-Šídák's *post hoc* test. ns = not significant. **b** Representative images of mutant hBEC colony formation in soft agar in donor 1. This analysis was repeated in the other two donors. Scale bars = 100 μm. **c** Quantification of soft agar colonies. Foci were counted as a colony if the circumference exceeded 50 μM. Mean +/- SD (n = 5 independent biological replicates). Mutants with >1 colonies are shown. Adj.P values were calculated by one-way ANOVA with multiple comparisons and Tukey's

*post hoc test*. ns = not significant. **d** Immunofluorescence staining of collagen-I invasion assays for the fibroblast marker vimentin and the epithelial cell marker EPCAM was used to distinguish pulmonary fibroblasts from invading epithelial cells. Left panel shows single channel images for vimentin (VIM, green), DAPI (DAPI, blue), and EPCAM (EPCAM, magenta). Right panel shows the number of EPCAM⁺ cells present in collagen. EPCAM⁺ cells were counted if ≥100 μm from collagen surface. Scale bars (single channels and merge) = 50 μm. Mean +/- SD of three replicates. Adj.P values were calculated by one-way ANOVA with multiple comparisons and Dunett's *post hoc* test. ns = not significant. Source data are provided as a Source Data file.

expansion (Fig. 4a). As expected, we validated an expansion of p63⁺ cells which was maximal in high-grade lesions (Fig. 4b, c). Using the same approach in ALI cultures, we found that p63⁺ cell expansion was maximal in TC + PKS in all three donors (Fig. 4d), strongly indicating that this mutant represents the highest disease grade. Expression of cytokeratins 5 and 6, also a LUSC biomarker, followed a trend similar to p63 (Supplementary Fig. 2b). We also identified inter-donor heterogeneity, most evident in donor 3, where p63⁺ cell expansion was as high in TC + KS as in TC + PKS. This is consistent with the results obtained when investigating anchorage-independent growth in the same mutants from donor 3 (Fig. 2c), which also showed similar levels of colony formation. Of note, expression of the lung adenocarcinoma marker TTF-1 was barely detectable in all mutants (Supplementary Fig. 3a).

Taken together, our analyses show that in the absence of lentiviral *SOX2*ᴼᴱ, mutant hBECs develop different alterations of the bronchial architecture, but do not acquire clear landmarks of transition to squamous lesions. *SOX2*ᴼᴱ in the TC background induces features of tumour initiation, including multi-layered morphology, anchorage-independent growth and squamous differentiation. In TC + PKS mutants, maximal expansion of p63-expressing cells, anchorage-independent growth and invasiveness indicate a transition to a malignant phenotype that requires SD, PI3K/Akt and OSR co-operation. Our observations also indicate a detrimental interaction between *SOX2*ᴼᴱ and PI3K/Akt pathway activation that is rescued by OSR pathway activation. This interaction and the requirement of the three pathways for malignant transformation could explain the co-occurrence of 3q-amplification (comprising *SOX2* and *PIK3CA*

amplification) and alterations targeting the OSR pathway in the classical LUSC subtype. In line with our results, interrogation of DepMap Consortium data (https://depmap.org/portal/)[16,17] revealed that LUSC was the most dependent on *NFE2L2* of all tumour types (Supplementary Fig. 3b, c). *NFE2L2* encodes for Nrf2, a master regulator of the OSR (or Nrf2) pathway that is regulated by *KEAP1* (Supplementary Fig. 3d). *SOX2* is also a co-dependency in *NFE2L2*-dependent cell lines, implying that *NFE2L2*-dependent cell lines are also dependent on *SOX2* (Supplementary Fig. 3e, f).

## Comprehensive gene-expression analysis of dysregulated pathways

Our combinatorial LUSC modelling approach allows us to deconvolve the effect of pathway dysregulation on gene expression. To do this, we carried out bulk RNA-seq of ALI cultures. Before focusing on the effect of specific pathways, we carried out PCA and WGCNA analyses to confirm that mutants from different donors behave similarly at the gene expression level. PCA revealed four clusters segregated primarily by genotype (*KEAP1* loss and *SOX2*ᴼᴱ) instead of donor (Supplementary Fig. 4a). As expected, TC + PS consistently clustered with wild type, TC and TC + P mutants. The role of *SOX2* and *KEAP1* in shaping LUSC transcriptomes is congruent with published studies[4,8,10]. WGCNA analysis (Supplementary Fig. 4b, c)[18] showed seven co-expression modules when the three donors were analysed globally (Supplementary Fig. 4b, Supplementary Data 1 and 2). Those high-stability modules were largely conserved when the donors were analysed individually (Supplementary Fig. 4c, Supplementary Fig. 5a–d).

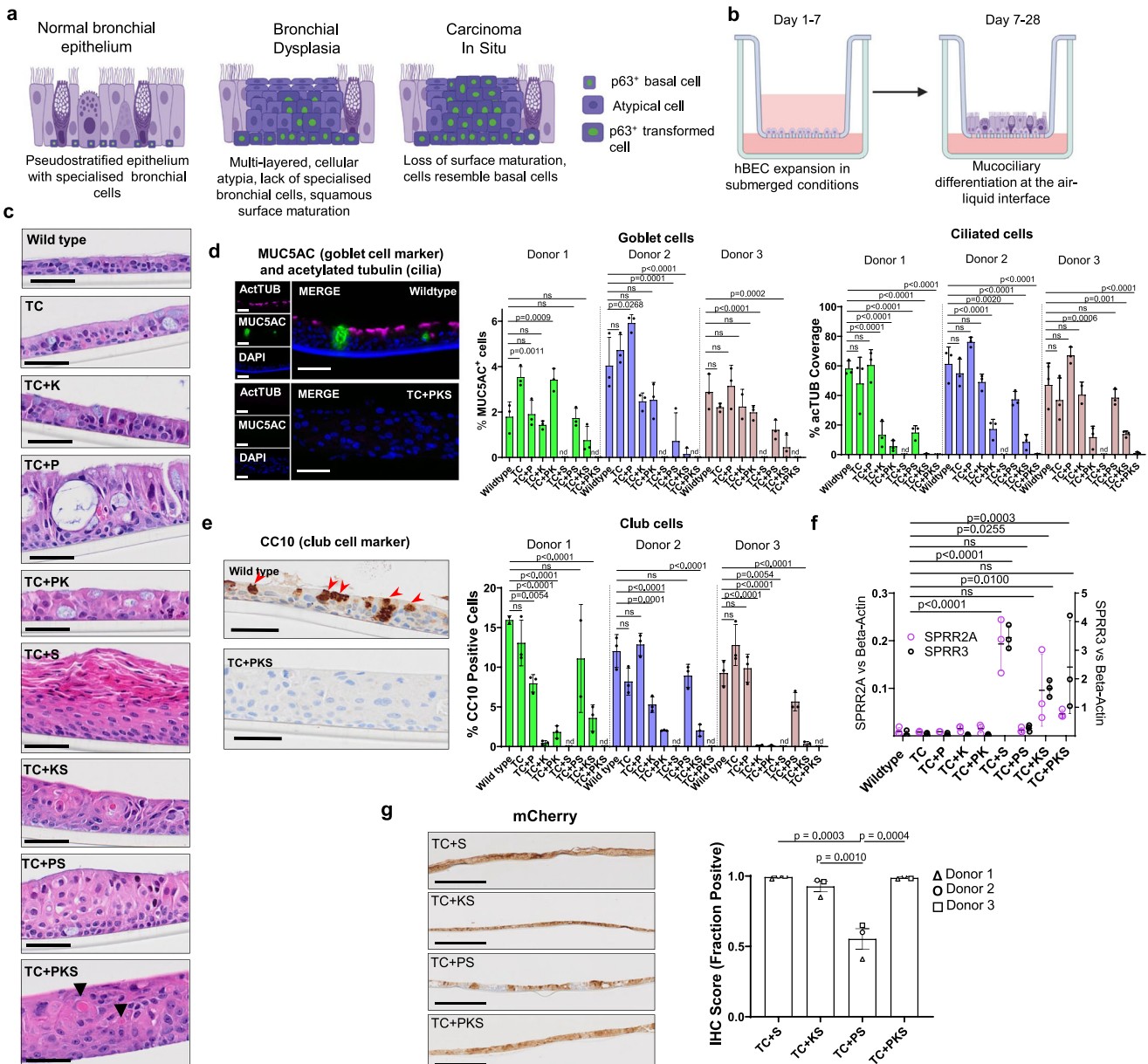

**Fig. 3 | Histological analysis of organotypic ALI cultures generated with mutant hBECs. a** Schematic of the developmental stages of LUSC progression. **b** Schematics showing the protocol to set up ALI cultures. **c** Histological sections of ALI cultures generated using mutant hBECs. Images are representative of three replicate assays carried out using hBECs derived from three donors. Arrowheads indicate areas of keratinisation. Scale bars = 50 μm. **d** Immunofluorescence of ALI cultures for MUC5AC (goblet cells, green), acetylated tubulin (cilia, magenta) and DAPI (nuclei, blue) to investigate changes in mucociliary differentiation. Left panel shows representative images of wild type and TC + PKCS mutants. Right panels show quantification of goblet cells shown as the % of MUC5AC-positive cells and acetylated tubulin positivity as the % cilia coverage. Mean +/- SD (*n* = 3 ALI cultures). Adj.P values were calculated by one-way ANOVA with multiple comparisons and Dunett's *post hoc* test. nd=not detected Scale bars = 50 μm. **e** Immunohistochemistry of ALI cultures for the club cell marker, CC10. Left panel shows examples of wild type and TC + PKCS mutants. Right panel shows quantification of the % CC10 positive cells. Mean +/- SD (*n* = 3 ALI cultures). Adj.P values were calculated by one-way ANOVA with multiple comparisons and Dunett's *post hoc* test. nd=not detected. Scale bars = 50 μm. **f** RT-qPCR showing the expression of two squamous differentiation markers in ALI cultures. Data shows the mean of three independent donors +/-SEM. Adj.P values were calculated by one-way ANOVA with multiple comparisons and Dunett's post hoc test. ns = not significant. **g** mCherry immunohistochemistry in *SOX2* overexpressing ALI cultures. Left panel shows representative images from one donor. Right panel shows quantification of mCherry-positive epithelium. Data is shown as the mean of three independent donors +/-SEM. Donor means were calculated from 3 replicates. Adj.P values were calculated by one-way ANOVA with multiple comparisons and Dunn's *post hoc* test. Figure 3a created in BioRender[53]. Figure 3b created in BioRender[54]. Source data are provided as a Source Data file.

After identifying limited heterogeneity between donors, we set out to analyse the transcriptomic data globally and identify differentially expressed genes regulated by each pathway relative to the TC background (Supplementary Data 3) followed by gene-set enrichment analysis (GSEA) (Fig. 5a). TC + PKS relative to TC were also interrogated to ensure gene-set enrichments attributed to one pathway is not suppressed by the concomitant dysregulation of additional pathways (Fig. 5a, Supplementary Data 3 and 4). Due to the size of the transcriptomic data, we focused on GO Biological Process and Hallmark collections. Positively enriched BP GO terms associated with *SOX2*^OE were dominated by epithelial development and keratinocyte biology (Fig. 5b and Supplementary Data 3 and 5), reflecting the transition to a

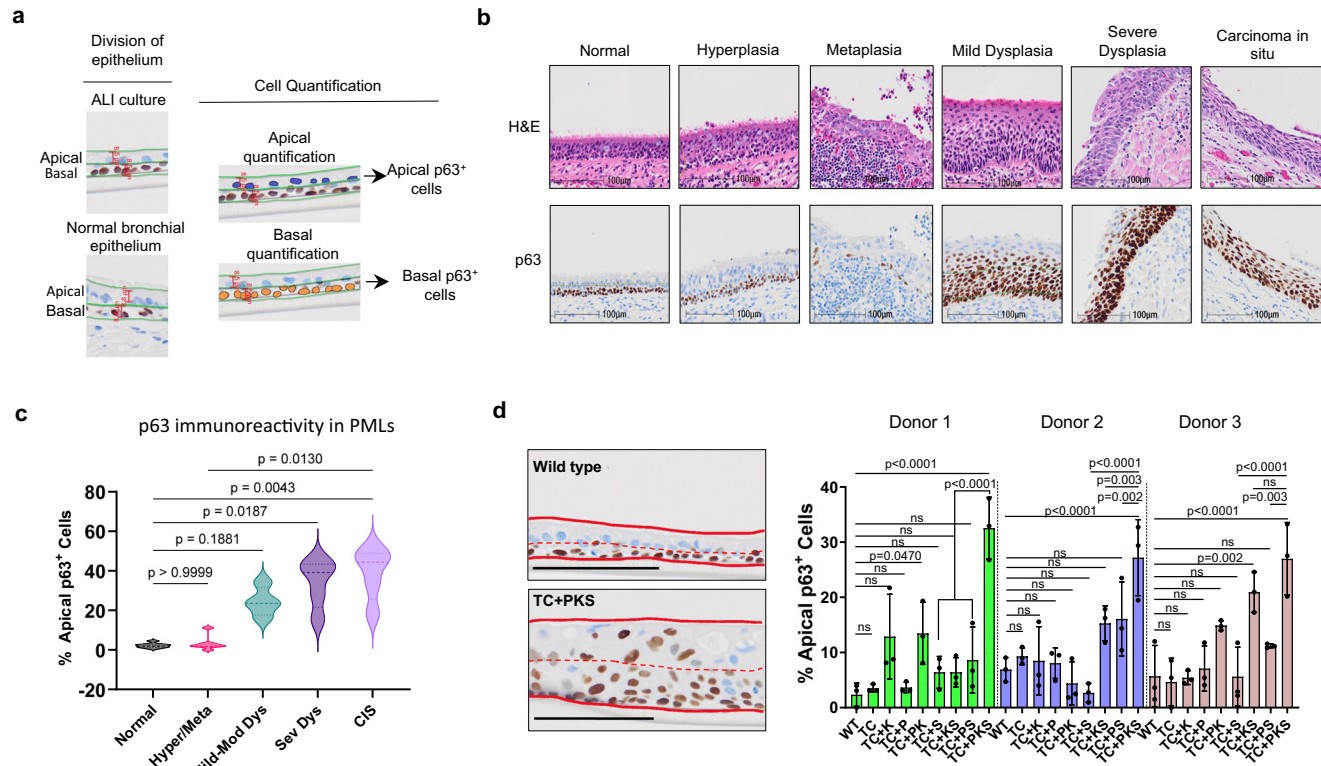

**Fig. 4 | Analysis of p63-expressing cell expansion in LUSC developmental stages and organotypic ALI cultures. a** Strategy for the quantification of apical and basal p63+ cells in clinical samples (right) and ALI cultures (left). Epithelia were split laterally into equal halves and separate analyses performed on each half. **b** Representative haematoxylin and eosin (top) and p63 immunohistochemistry (bottom) staining of LUSC developmental stages obtained from clinical samples. This analysis was repeated in the three donors. **c** The % of p63 positive cells which reside in the apical half of the airway epithelium is significantly higher in high-grade premalignant lesions than in normal airway epithelium. Analysis used eight clinical samples with different spectrums of premalignant lesions (normal epithelium = 8, hyperplasia/metaplasia = 7, mild-moderate dysplasia = 4, severe dysplasia = 5 and carcinoma in situ = 5). Significance was calculated using Kruskal-Wallis test with multiple comparisons and Dunn's *post hoc* test. **d** TC + PKS mutant hBECs generate ALI cultures with a higher proportion of p63+ apical cells. Left panel shows representative immunohistochemistry images of wild type and TC + PKS ALI cultures stained for p63. Solid red lines indicate top and bottom of epithelium and dashed red lines mark the lateral division of the epithelium into its apical and basal halves. Scale bars=100 μm. The quantification of % p63+ apical cells (right panel) in ALI cultures revealed a significant increase in TC + PKS mutation. Mean +/- SD (n = 3 ALI cultures). Adj.P values were calculated by one-way ANOVA with multiple comparisons and Holm-Šídák's *post hoc* test. ns = not significant. Source data are provided as a Source Data file.

squamous keratinised epithelium, characteristic of squamous meta-plasia. Genes upregulated in TC + S and TC + PKS mutants were also enriched in RAS-RTK pathway associated GO terms (Fig. 5b, and Supplementary Data 3 and 5) and in the KRAS SIGNALLING UP hallmark (Fig. 5c, Supplementary Fig. 6a, Supplementary Data 4 and 5). RAS-RTK signalling in LUSC has been functionally linked to *SOX2* in cooperation with *TP63* to induce *EGFR* expression[19] and with the expression of EGFR ligands different from EGF[8]. We confirmed that *SOX2*[OE] upregulated *TP63* and EGFR ligands, but not EGFR. (Fig. 5d and Supplementary Data 3). We identified a heterogeneous group of positively enriched GO terms related to protein processing, including multiple serine protease inhibitors (Fig. 5b, e, Supplementary Fig. 6b and Supplementary Data 3 and 5). Among these was *PI3*, which encodes a neutrophil-elastase inhibitor and known component of the cornified envelope[20], highly upregulated in *SOX2*[OE] mutants (Supplementary Fig. 6c, Supplementary Data 3). Therefore, we hypothesised that *PI3* might counteract the known tumour suppressive activity of neutrophil-elastases to promote LUSC progression[21]. A significant positive correlation between *SOX2* and *PI3* mRNA (Spearman=0.24 q-value = 0.0335) and protein (Spearman=0.268, q-value = 0.0214) expression was confirmed using CPTAC data[8], but not the TCGA cohort[9]. When we interrogated a transcriptomic database of pre-malignant lesions previously published[22,23], maximal *PI3* expression was observed in high-grade and invasive carcinomas (Fig. 5f), implying a function in the transition to invasive carcinomas.

As expected, processes downregulated following *SOX2*[OE] largely related to cilia biology (Fig. 5g, Supplementary Data 3 and 5), reflecting bronchial-squamous reprogramming (Fig. 3c–f). A cluster of negatively enriched ontologies related to protein transport contained gene-sets consisting of MHC-II subunits (Fig. 5g-i, Supplementary Fig. 6d, Supplementary Data 3 and 5). Two recently published articles that analyse premalignant lesions[24] or use LUSC human and mouse models[25] reported the downregulation of epithelial MHC-II expression mediated by the MHC-II transcriptional activator, *CIITA*. However, our data showed that *CIITA* was downregulated by *SOX2*[OE] (Fig. 5i and Supplementary Data 3). Furthermore, we confirmed a negative correlation between *CIITA* expression and MHC-II genes in LUSC with *SOX2* expression using both the TCGA and CPTAC cohorts (Supplementary Fig. 6e, f). Our analysis of SOX2-associated expression data strongly indicates that *SOX2* amplification in LUSC is multi-faceted and con-stitutes a hub of pro-tumourigenic cues, not limited to cell autono-mous processes (squamous differentiation, RTK-RAS pathway activation), but also likely to remodel the immune microenvironment.

Consistent with the function of the OSR pathway, *KEAP1* inacti-vation resulted in positive enrichment of redox metabolism and xenobiotic detoxification gene-sets (Fig. 6a, and Supplementary Data 6). A similar set of significantly enriched terms has been identified in the classical LUSC subtype, highlighting the importance of this pathway in shaping LUSC transcriptomes[4,8,10]. Further to this, we observed an enrichment of amino acid transport gene-sets known to

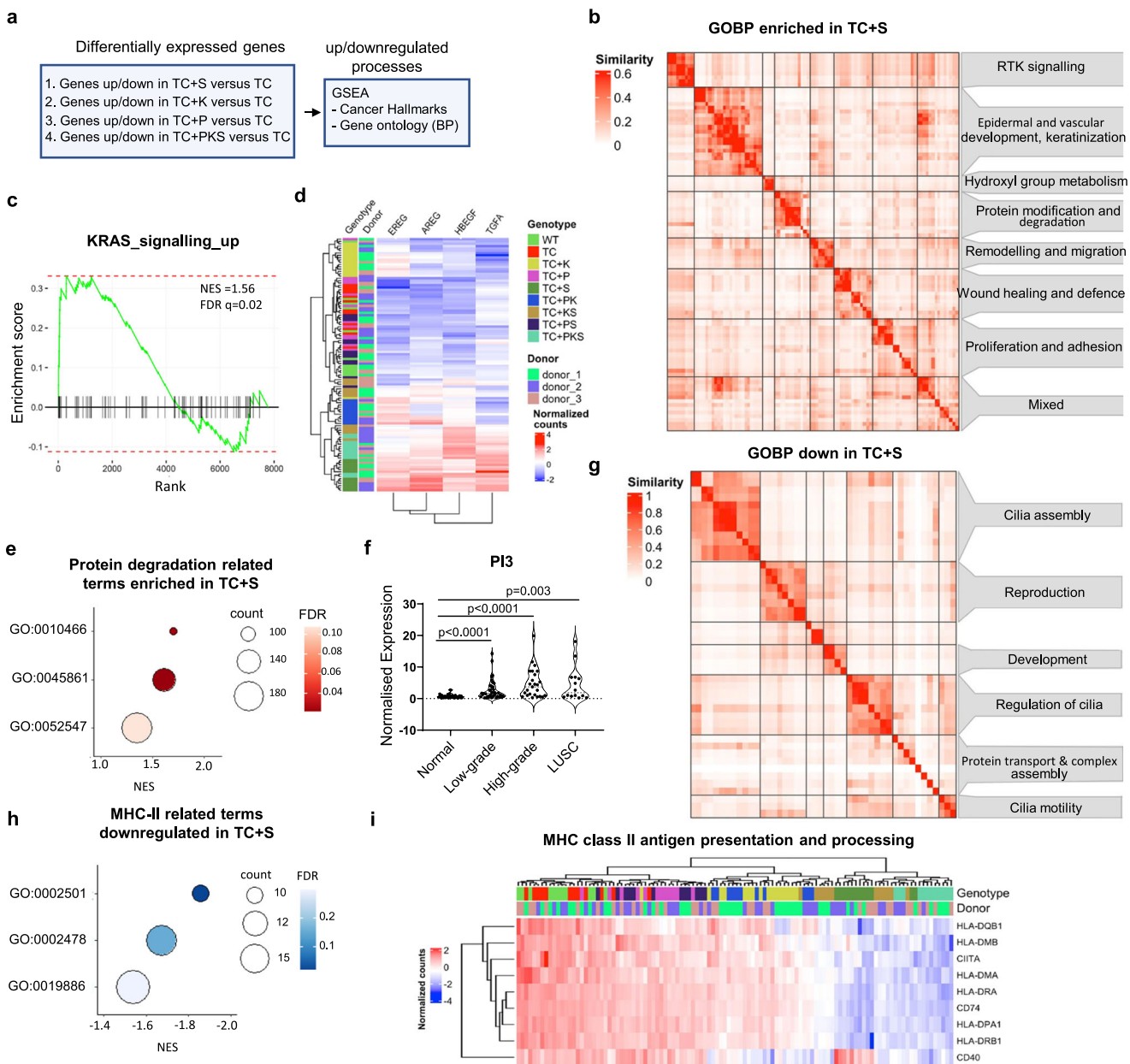

**Fig. 5 | Global transcriptomic effects of *SOX2*^OE in mutant hBECs. a** Strategy for the identification of processes associated with each single pathway (TC + S/K/P) and all pathways in combination (TC + PKS). Gene expression changes associated with the dysregulation of each single pathway and all pathways in combination were identified by carrying out differential gene expression (TC + P/S/K/PKS versus TC) followed by gene set enrichment analysis (GSEA). **b** Clustergram of significant positively enriched gene ontologies (Biological Process) using GSEA of genes differentially expressed in TC + S versus TC comparisons (q-value ≤ 0.05). Terms were aggregated by semantic similarity using the simplifyEnrichment package. **c** GSEA plots showing the significant positive enrichment (q-value < 0.05) of the KRAS signalling up hallmark (MSigDB, H: hallmark gene sets) in genes differentially expressed in TC + S versus TC (q-value ≤ 0.05). **d** Heatmap showing the normalised expression of EGFR ligands across the entire sample set. **e** Dot plot showing positively enriched gene ontologies related to protein degradation within the 'protein modification and degradation' cluster from (b). GO:0010466=negative regulation of peptidase activity, GO:0045861=negative regulation of proteolysis, GO:0052547=regulation of peptidase activity. **f** Plot showing the expression of *PI3* in normal (n = 42), low (n = 41) and high-grade (n = 25) premalignant lesions and LUSC (n = 14). Significance was calculated by comparing each group to 'normal' using the two-sided Mann-Whitney test. GEO accession: GSE33479. **g** Clustergram of significant (q-value < 0.05) negatively enriched gene ontologies (Biological Process) using GSEA of genes differentially expressed in TC + S versus TC comparisons (q-value ≤ 0.05). **h** Dot plot showing negatively enriched gene ontology terms related to MHC-II peptide presentation within the 'protein transport and complex assembly' cluster from (g). GO:0019886=antigen processing and presentation of exogenous peptide antigen via MHC class II, GO:0002478=antigen processing and presentation of endogenous peptide antigen, and GO:0002501=peptide antigen assembly with MHC protein complex. **i** Heatmap showing the normalised expression of genes involved in MHC-II antigen processing and presentation across the entire sample set. Source data are provided as a Source Data file.

be bona-fide Nrf-2 targets (Fig. 6a-c, Supplementary Fig. 7a and Supplementary Data 3, 4 and 6). We confirmed upregulation of a subset of these transporters in LUSC tumours with OSR-targeting alterations; *SLC1A4* and *SLC1A5* (glutamine influx), *SLC7A11* (cystin influx), *SCL75A* (essential amino acids), *LRRC8D* (anion channel) and *SLC3A2* (ancillary

to *SLC7A5* and *SLC7A11*) (Fig. 6d). To the best of our knowledge, *LRRC8D* has not previously been reported as an Nrf-2 target but has been linked to chemotherapy resistance[24]. Finally, we observed an enrichment in the *glucose-6-phosphate metabolic process* GO term (lipid and glucose metabolism cluster) (Fig. 6a, e, and Supplementary

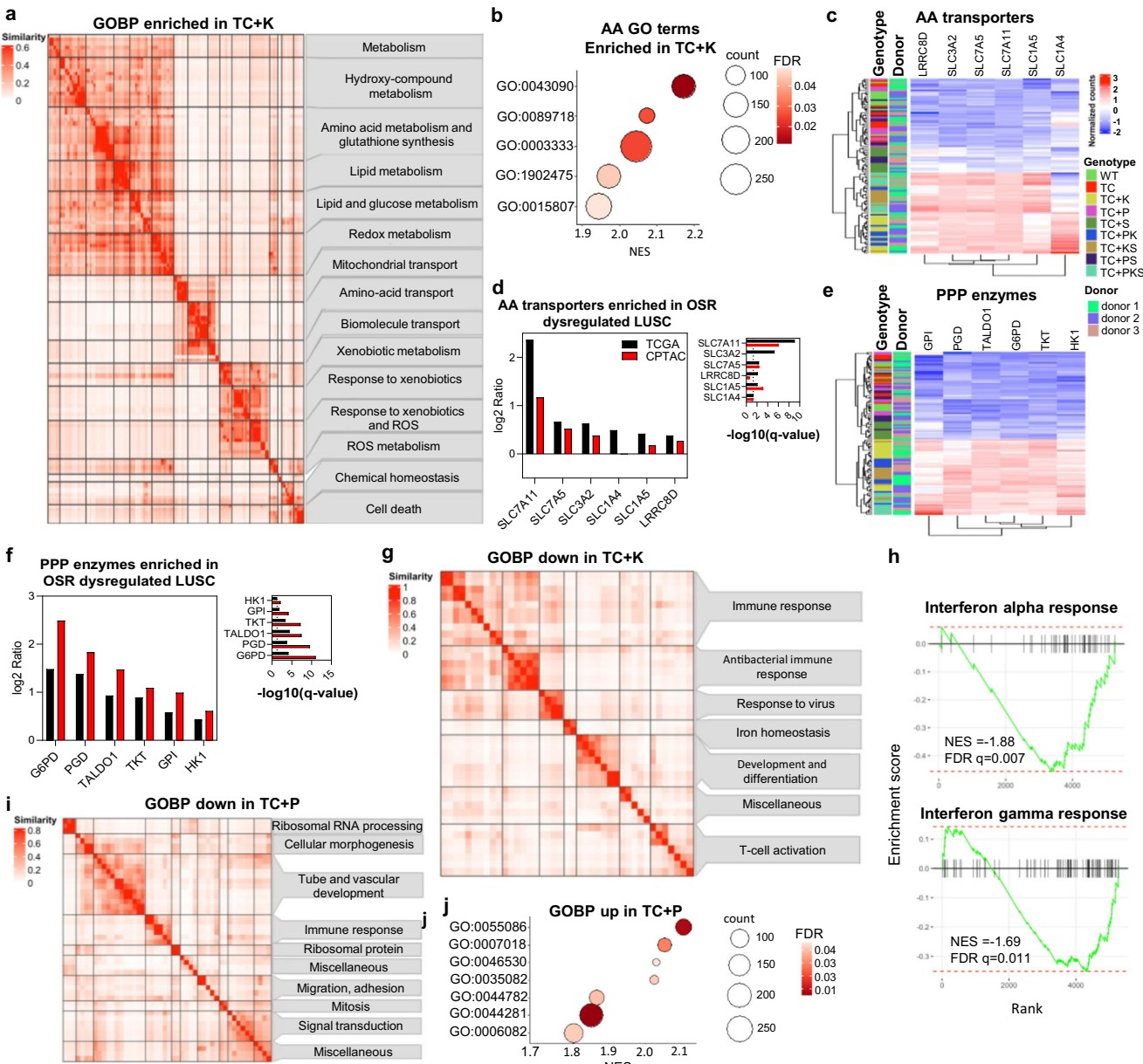

**Fig. 6 | Global transcriptomic effects of *KEAP1* and *PTEN* inactivation in mutant hBECs. a** Clustergram of significant positively enriched gene ontologies (GO Biological Process) using GSEA of genes differentially expressed in TC + K versus TC comparisons (q-value < 0.05). **b** Dot plot showing positively enriched gene ontology terms related to amino acid transport within the 'amino acid transport' cluster from (a). GO:0015807=amino acid transport, GO:0003333 = amino acid transmembrane transport, GO:1902475=alpha amino acid transmembrane transport, GO:0089718=amino acid import across plasma membrane, and GO:0043090=amino acid import. **c** Heatmap showing the normalised expression of amino acid transporters across the entire sample set. **d** Bar chart showing Log2Ratios comparing the expression of amino acid transporters between LUSC tumours with and without OSR pathway alterations (*KEAP1*, *NFE2L2*, and *CUL3*) in 179 LUSC samples from the TCGA (mRNA) and 80 LUSC samples from CPTAC (protein). Left panel shows Log2Ratios. Right panel shows -log10(q-value). Dotted line indicates significance threshold (q-value < 0.05). p-values were derived from two-sided student's t-tests and q-values from Benjamini-Hochberg procedure. **e** Heatmap showing the normalised expression of pentose phosphate pathway (PPP) enzymes in the entire sample set. **f** Bar chart showing Log2Ratios comparing the expression of PPP enzymes between LUSC tumours with and without OSR

pathway alterations as shown in (d). Left panel shows Log2Ratios. Right panel shows -log10(q-value). Dotted line indicates significance threshold (q-value < 0.05). p-values were derived from two-sided student's t-tests and q-values from Benjamini-Hochberg. **g** Clustergram of significant (q-value < 0.05) negatively enriched gene ontologies (GO Biological Process) using GSEA of genes differentially expressed in TC + K versus TC comparisons (q-value < 0.05). **h** GSEA showing the significant negative enrichment (q-value < 0.05) of the response to interferon alpha (top) and gamma (bottom) hallmarks (MSigDB, H: hallmark gene sets) in genes differentially expressed in TC + K vs TC (q-value < 0.05). **i** Clustergrams of negatively and positively enriched gene ontologies (GO Biological Process) (q-value < 0.05) in GSEA comparing TC + P versus TC (q-value < 0.05, log2FC). **j** Dot plots showing the seven positively enriched gene ontologies (GO Biological Process) from genes differentially expressed in TC + P versus TC comparisons (q value < 0.05). GO:0055086=nucleobase-containing small molecule metabolic process, GO:0007018=microtubule-based movement, GO:0046530=photoreceptor cell differentiation, GO:0035082=axoneme assembly, GO:0044782=cilium organisation, GO:0044281=small molecule metabolic process, GO:0006082=organic acid metabolic process. Source data are provided as a Source Data file.

Data 3, 4 and 6) that included most pentose-phosphate pathway (PPP) enzymes. These enzymes are known Nrf-2 targets and are upregulated in LUSC with OSR alterations (Fig. 6f). PPP upregulation via OSR activation contributes to redox homoeostasis by maintaining high GSH/GSSG ratios and NADPH production, although other redox metabolism unrelated functions, such as nucleotide biosynthesis, could play an important role.

*KEAP1* inactivation drove the downregulation of immunity-related gene-sets (Fig. 6g and Supplementary Data 6) possibly reflecting the negative enrichment for interferon response hallmarks (Fig. 6h and Supplementary Data 6) and the enrichment of both hallmarks that we observed in module 6 genes (downregulated *KEAP1* mutants) of our WGCNA analysis (Supplementary Fig. 4b and 7b). This immunosuppressive role through interferon response inhibition by OSR has been reported in lung adenocarcinoma but not in LUSC[25]. Whilst TC + PKS mutants showed similar results, they did not reach significance for the interferon gamma response hallmark (Supplementary Fig. 7c). Likewise, tumours with OSR-targeting alterations also showed downregulation of genes in interferon and inflammation related hallmarks (Supplementary Fig. 7d, Supplementary Data 2), and suppression of interferon responses was most prominent in the classical subtype[8]. Therefore, our model therefore recapitulates a wide range of reported metabolic and immune-related transcriptional changes mediated by OSR activation.

PI3K/Akt activation by *PTEN* truncation resulted in a more limited transcriptional response than *SOX2*OE and *KEAP1* truncation (Supplementary Fig. 7e and Supplementary Data 3). This is consistent with our PCA and WGCNA analysis (Supplementary Fig. 4a, b) showing that the latter alterations are the most determinant factors in shaping transcriptomes, and on the other hand, with reports showing that LUSC subtypes are not determined by PI3K/Akt-targeting alterations[8,10]. GSEA analysis showed the expected positive enrichment in the MTORC1 and Glycolysis hallmarks (Supplementary Data 7) and predominantly negative enrichment in GO terms related to multiple processes, such as ribosome biology, migration and development (Fig. 6i, j, Supplementary Data 7).

As expected, we identified a significant overlap between the GO terms enriched in patients with SOX2 amplification and the enrichment in the TC + S vs TC comparison and samples with alterations in the OSR pathway and the TC + K vs TC comparison (Supplementary Fig. 8a). No significant overlap between the enrichment in samples with *PTEN* alterations and the TC + P vs TC comparison. This is likely to be caused by the more limited number of genes and GO terms enriched in these patients, which limits the statistical analysis.

Finally, we hypothesised that if our model is indeed recapitulating the classical LUSC subtype, the enrichment scores for the GO Biological Process terms perturbed by the three pathways acting concomitantly (TC + PKS vs TC) should correlate positively with the same terms in the classical subtype. As expected, the enrichment scores for those terms in the classical subtype were positively correlated with the TC + PKS mutant, (Supplementary Fig. 8b), confirming that the similar processes are altered. Interestingly, the correlation was also positive for the EMT-E subtype, but the high level of fibroblast infiltration and low tumour purity in this subtype (Supplementary Fig. 8b, c)[8] are important limitations for the interpretation of this result.

## An evolutionary sequence of pathway activation in LUSC development

Although the main aim of our combinatorial strategy of genetic manipulation was to unravel how the most relevant LUSC pathways cooperate to drive LUSC progression, our results can be mined to hypothesise the most likely trajectory of pathway activation. Our morphological analysis revealed that *SOX2*OE abrogates bronchial epithelial architecture and induces SD (Fig. 3c–e, and Supplementary Fig. 7a, b), two processes that occur early in LUSC development,

whereas additional *KEAP1* and *PTEN* inactivation drive further p63-positive cell expansion and invasiveness (Figs. 2d, 4d). Therefore, *SOX2* amplification, or more generally, activation of the SD pathway, might occur earlier than OSR and PI3K/Akt pathways (Fig. 7a). Additionally, our observation that *KEAP1*-loss rescues the negative interaction between *SOX2*OE and *PTEN* truncation suggests that PI3K/Akt pathway activation might occur at an evolutionary stage that avoids the detrimental interaction with SOX2 (Fig. 7a). Therefore, according to our results, the sequence SD-OSR-PI3K/Akt is the most likely evolution of pathway activation according to our results (Fig. 7a). We next ascertained whether this hypothetical evolutionary trajectory is consistent with observations in human specimens. Published LUSC evolutionary histories[26] lack the temporal resolution required for this analysis. Moreover, the lack of genomic characterisations of developmental LUSC stages prevents us from inferring an evolutionary sequence of somatic events. However, the preinvasive lesion cohort generated by Mascaux and colleagues[22] can be interrogated to map the onset of pathway dysregulation using transcriptional signatures. Thus, we carried out ssGSEA analysis on the latter data using gene-sets associated with the three pathways (Fig. 7b–d). As expected, an increase in squamous differentiation-related enrichment scores was significant in squamous metaplasia and increased OSR- and PI3K/Akt-related scores became significant in squamous metaplasia and moderate dysplasia respectively (Fig. 7c, d). These observations support that PI3K/Akt pathway activation occurs later than SD and OSR pathways, which appear to be upregulated simultaneously. Though stromal contamination, LUSC molecular subtypes and the assumption that LUSC development is homogeneous across patients might be confounding factors, these results indicate that the onset of PI3K/Akt signalling occurs later in LUSC evolution, perhaps due to a reduction in cell fitness caused by the interaction between PI3K/Akt and *SOX2* gain-of-function.

## Investigating drug responses in LUSC using the in vitro model

Testing drugs using systems that recapitulate the native intra-epithelial architecture constitute a more physiological system than submerged cultures. However, the experience in ALI cultures to measure responses to cancer treatments is very limited. To test whether our system allows the measurement of read-outs that correlate with anti-cancer drug responses, we treated wild type and TC + PKS ALI cultures with three concentrations of cisplatin, a drug used to treat advanced LUSC (Supplementary Fig. 9a) to confirm dose-dependent and genotype-dependent responses. This design facilitates the comparison of effect in donor-matched normal and cancerous bronchial epithelia, as the TC + PKS mutant demonstrates the most invasive phenotype. To quantify the effect of cisplatin, we multiplexed three read-outs: cell viability (resazurin reduction), apoptosis (caspase 3/7 activity) and lactate dehydrogenase (LDH) release to basal media, a parameter correlated with cell death.

LDH activity at 72 h showed a dose-dependent cytotoxic effect in both genotypes and a more intense effect in the TC + PKS mutant (Supplementary Fig. 9b). We also confirmed a cumulative cytotoxic effect over time that was dose-dependent with no significant effect in the 5 µM concentration (Supplementary Fig. 9c). Cell viability did not show the dose dependent effect in the TC + PKS mutant at 72 h (Supplementary Fig. 9d) and a marginal increase was observed at 20 µM and 100 µM concentration. Caspase 3/7 activity at 72 h exhibited a dose-dependent increase consistent with the LDH activity results (Supplementary Fig. 9e). The unexpected results for cell viability in the TC + PKS mutant could be caused by a higher basal level and release of reductases after cell death that mask a drop in viability. Additionally, the lack of a viability readout hinders a proper normalisation of the caspase 3/7 activity data.

In summary, LDH activity in basal media and caspase 3/7 provide a good and flexible read-out of cytotoxicity in our LUSC model.

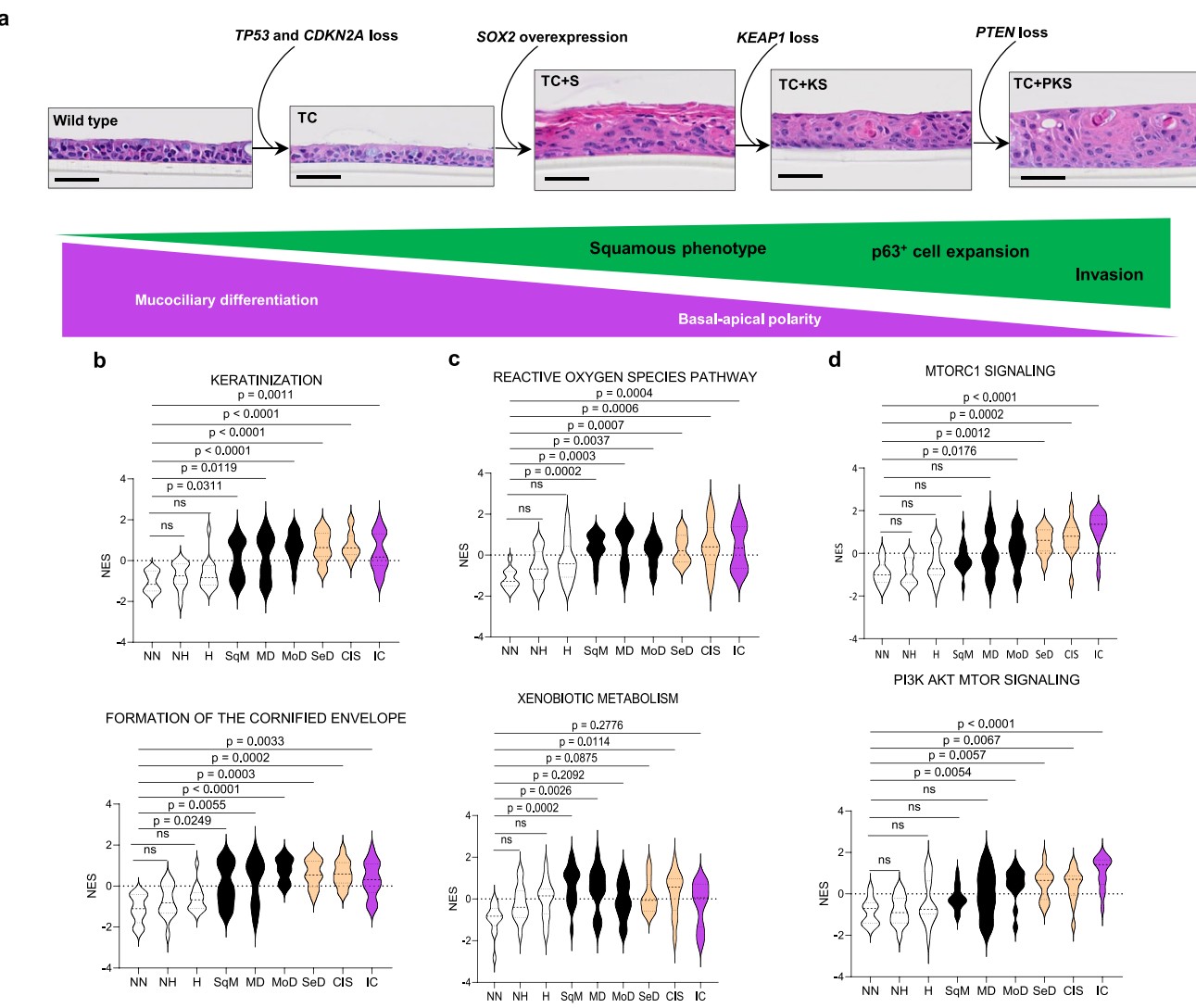

**Fig. 7 | A hypothetical evolutionary sequence of pathway activation in LUSC development. a** Hypothesised evolutionary sequence of pathway activation in LUSC development based on the morphological and molecular characteristics of our model including loss of mucociliary differentiation and epithelial polarity, and gain of a squamous phenotype, expansion of p63+ cells, and acquisition of an invasive phenotype. Scale bars = 50 μm. This evolutionary sequence was confirmed in the three donors (see Fig. 3c and Supplementary Fig. 2a). **b**–**d** Normalised enrichment scores (ssGSEA) calculated from the dataset published by Mascaux et al. (2019) (GEO accession: GSE33479) for gene signatures associated with activation of the squamous differentiation pathway (**b**) including keratinisation (GO:0031424, top) and formation of the cornified envelope (Reactome R-HSA-68009371, bottom), the OSR pathway (**c**) including the reactive oxygen species pathway (Hallmark M5938, top) and xenobiotic metabolism (Hallmarks M5934, bottom), and the PI3K/AKT pathway (**d**) including MTORC1 signalling (Hallmarks, M5924, top) and PI3K AKT MTOR signalling (Hallmarks M5923, bottom). Scores are shown for each stage of LUSC development including normal (NN), normal hypofluorescent (NH), hyperplastic (H), squamous metaplasia (SqM), mild dysplasia (MD), moderate dysplasia (MoD), severe dysplasia (SeD), carcinoma in situ (CIS), lung squamous cell carcinoma (LUSC). White = normal, black = low grade, green = high grade, and red = invasive carcinoma. Significance (q-value < 0.05) was calculated using Kruskal-Wallis testing with multiple comparison and Dunn's *post hoc* test. Source data are provided as a Source Data file.

However, resazurin-based assays are not a suitable system to measure cell viability in our system and alternatives should be considered.

## Discussion

Advances in translational cancer research rely on the availability of tractable models that recapitulate key cancer traits. Mouse models are valuable for understanding basic LUSC biology. However, the availability of resources and inter-species differences hinder the modelling of increasingly complex genotypes, evolutionary trajectories and molecular subtypes. We demonstrated here that genetically engineered hBECs constitute a tractable, biologically relevant system to study LUSC as demonstrated by the expression of p63 and cytokeratins 5/6, absence of TTF-1, and the presence of keratinisation. When compared to previous articles reporting LUSC models using hBECs, our approach is more ambitious. We implemented our strategy with the same cell type from three donors without cancer or airway disease to ensure reproducibility. Additionally, previous approaches involved a limited number of genetic manipulations or utilised immortalised cells[27,28].

Another emerging in vitro approach in LUSC is patient-derived organoids (PDOs)[29–31]. Whereas LUSC PDOs are promising tools as preclinical models, our approach has intrinsic advantages. It attempts to build an isogenic system to capture commonalities among patients with similar genomic profiles. This advantage can simplify preclinical research, as obtaining and growing PDOs with similar genomic profiles is costly in time consuming. Additionally, LUSC PDOs typically exhibit short-term growth, whereas the versatility of our approach allows an

expansion of cell stocks that can be adapted to any research requirements.

However, the versatility of our system for basic LUSC research is its most prominent advantage. It enables a detailed functional dissection of the processes that drive LUSC. Future research could include but is not limited to the investigation of epistatic interactions between mutations, early disease biology, and predisposition and risk factors by manipulating hBECs from high-risk populations (namely smokers, COPD patients or genetically predisposed populations).

Although both PDOs and our genetic human model can be used orthotopically or co-cultured with extra-epithelial components in vitro to incorporate TME components, only PDOs can be co-cultured with tumour-instructed immune cells. However, the use of humanised mouse models that combine our hBECs system with allogeneic immune cells can compensate this disadvantage.

Similar models developed by genetic manipulation of primary human cells published for other cancer types made use of only one donor in a melanoma model[32], or cells isolated from multiple patients with colorectal cancer[33] and cells from spatially separated intestinal regions[34] in colorectal cancer models. Overall, our cell biology results indicated that concomitant activation of the SD, PI3K/Akt and OSR pathways was required for malignant transformation. Although some inter-donor heterogeneity was detected, a holistic vision of our results revealed homogeneous phenotypes in the three donors who were of similar age, non-smokers and lacked airway disease.

We conclude that our approach has generated a human model of the classical LUSC subtype[4,8] as [1] the three pivotal pathways investigated were required for invasive transformation which is consistent with the accumulation of OSR pathway mutations and 3q-amplification (*SOX2* and *PIK3CA*) in classical LUSC subtype[8], [2] OSR activation was required for the expansion of *SOX2*$^{OE}$ cells with concomitant PI3K/Akt activation, and [3] OSR and *SOX2* downregulated pathways involved in tumour immunity, which could contribute to the 'immune-cold' environment observed in the classical LUSC subtype. A global gene-expression correlation analysis confirmed a positive correlation of the invasive mutant (TC + PKS) with the classical subtype and, interestingly, the EMT-E subtype[8]. Whereas 3q-amplification and Nrf-2 pathway alterations are not excluded from the EMT-E subtype, they are certainly less frequent in this subtype. Although, in principle, this observation supports that our model also recapitulates the EMT-E subtype, we need to be cautious with this conclusion as the high levels of fibroblast contamination reported for this subtype[8] makes this result difficult to interpret. Mouse models have not demonstrated clearly the requirement for SD, PI3K/Akt and OSR activation in defining classical LUSC subtype. Ferone et al. (2017) used a strategy that involved *Sox2*$^{OE}$ and *Pten*$^{ko}$ without targeting the OSR pathway which successfully resulted in tumourigenesis. Notwithstanding inter-species differences and the lack of evidence of invasion in the latter model, positive selection for cells with OSR activation during tumour development, or potential similarities with non-classical subtypes in this model might explain this difference. More recently, Pan and colleagues[35] reported differences in LUSC driven by *Kmt2d* and *Pten* using GEMMs (red). This study clearly revealed that both genes can drive LUSC after a long latency period and different sensitivities to RTK-RAS pathway inhibition. However, it is unclear what LUSC subtypes these models are representing.

Our observations suggest new translational perspectives. For example, our results indicate that tumours with 3q-amplification and OSR-targeting alterations are likely to be more sensitive to experimental therapies targeting the Nrf-2 pathway (e.g. glutaminase inhibitors) and that rescuing the negative interaction between SOX2 and PI3K/Akt activation might constitute an elusive basic function of OSR activation in LUSC. A likely mechanism could be OSR mediated detoxification of endogenous toxic ROS originating from cumulative *SOX2* and PI3K/Akt driven metabolic reprogramming[36,37] or increased metabolic demands caused by concomitant *SOX2*$^{OE}$ and PI3K/Akt, such as a higher demand for amino acids and glutaminolysis reliance, that are compensated by upregulation of essential amino acid (*SLC7A5*) and glutamine (*SLC1A5*) transporters. Our results might explain why lung cancer prevention strategies based on the use of antioxidants increase the risk of lung cancer, as these agents mimic Nrf-2 pathway activation and might favour the expansion of clones with 3q-amplification[36,37].

Therapies that target the OSR pathway by limiting glutamate biosynthesis for glutathione production have been tested in trials (NCT04265534 and NCT04471415) without positive results. Addressing this lack therapeutic effect and testing new strategies to target the OSR pathway can be approached with our human model as it has recapitulated the wide range of gene expression changes previously reported for the OSR pathway, and additional changes, reflecting a complex interplay between interdependent and/or functionally redundant amino acid transporters.

We also show regulation of transcriptional programmes related to acquired and innate immunity by the SD and OSR pathways. SOX2 downregulates the neutrophil-elastase inhibitor *PI3* and MHC-II while OSR activation lowered type-I and II interferon responses. Although further research is needed to assess the functional consequences of those changes in immunosurveillance, our observations support previous studies that have already reported an immunomodulatory function for SOX2, although with different mechanisms[38].

Notwithstanding that potential strategies to target the individual cell autonomous and TME-related processes mediated by SOX2 must be tested[21,25], the multi-faceted SOX2 oncogenic functions emphasise the need to target SOX2 itself. New therapeutic tools such as PROTACs or approaches analogous to Omomyc could facilitate this endeavour[39]. Similarly, OSR inhibition is likely to enhance type-I and II interferon responses and remodel the immune cell landscape of LUSC. This is in line with the limited therapeutic response to immunotherapies observed in cancer patients and mouse models with OSR alterations[25,40,41]. Our human LUSC model also provide an enhanced experimental tool to investigate the latter strategies in LUSC.

We have provided evidence that supports the use of the human LUSC model in preclinical research and the importance of using multiple read outs. ATP quantification is a potential alternative to resazurin systems to determine cell viability, but it requires cell lysis and therefore, multiplexing with caspase 3/7 activity is not possible.

Finally, the comparison of our observations with transcriptomic signatures in patient samples follows an evolutionary trajectory in which activation of SD and OSR programs precede PI3K/Akt activation, but with several caveats. This trajectory is at odds with *SOX2* and *PIK3CA* co-amplification in 3q26, which suggests simultaneous SD and PI3K/Akt activation. However, gene-dose to phenotype correlations likely differ amongst genes, as might the timing of phenotype emergence following a copy number event. In fact, 3q26 amplification is detectable in progressive squamous metaplasias[42] and increases during LUSC development[43]. If *PIK3CA* gene dose is less efficient in inducing the pathway, this would result in an evolutionary 'delay' of PI3K/Akt activation.

We set out to provide an informative, tractable in vitro LUSC model as an alternative, or an animal use reducing prelude, to more expensive, technically challenging and animal consuming in vivo modelling of LUSC. At the moment, our model focuses primarily on cell-intrinsic effects without the complexity of the native TME, which constitutes a limitation. However, our model incorporates intra-epithelial conditions and can be progressively upgraded to incorporate TME components. Our model can now be applied to address further basic and translational research questions, the most obvious being a reverse genetic approach to model LUSC heterogeneity using comparative genomic studies. For instance, our system can be used to test whether other drivers of the three LUSC prevalent pathways phenocopy the alterations used in this study and trigger the same

vulnerabilities or, alternatively, to assess the function of unrelated drivers. Our model also provides an opportunity to explore the biology of risk factors, including smoking and COPD by manipulating hBECs derived from appropriate donors. Although the study of metastasis is an additional limitation of our model, the use of bioengineering solutions, such as organ-on-a-chip technology, might overcome this limitation. Lastly, the identification of new biomarkers for LUSC prevention and early detection methods, drug resistance, and immune-oncology studies are examples of the multiple translational opportunities to exploit in our human LUSC model.

## Methods

### Study approval
This research complies with all the ethical regulations of the University of Manchester and the Cancer Research UK Manchester Institute. Namely, patient samples were obtained from the Manchester Cancer Research Centre (MCRC) Biobank (Approval #23_CALO_01). The MCRC Biobank holds a generic ethics approval (Ref: 18/NW/0092) which can confer this approval to users of banked samples via the MCRC Biobank Access Policy. Banked samples were obtained with full, informed consent of the donor and released in accordance with the consent under which it was obtained. Genetic manipulation was carried out under approval by Genetic Modification and Biohazards Safety Committee (University of Manchester).

### Cell culture
Mouse embryonic 3T3-J2 fibroblasts (Kerafast, EF3003) were cultured in DMEM (Gibco™, 11995065) with 10% bovine serum (Fisher Scientific, 16-107-078), 2 mM GlutaMAX™ Supplement (Gibco™, 35050061) and 5% Penicillin-Streptomycin (Gibco™, 15070063). Lenti-X 293 T cells (Takara, 632180) and primary human lung fibroblasts (Lonza Bioscience, ID: HLF-2, batch:20TL356516) were cultured using DMEM (GibcoTM, 11995065) with 10% foetal bovine serum (LabTech, 80837), 2 mM GlutaMAX™ Supplement (GibcoTM, 35050061) and 5% Penicillin-Streptomycin (GibcoTM, 15070063). Normal human bronchial epithelial cells (2 males and 1 female based on donor self-report) were purchased from Lonza (CC-2540S) and cultured using the 3T3-J2 feeder layer co-culture system described by Hynds et al.[11]. Sub-confluent mouse embryonic 3T3-J2 fibroblasts were mitotically inactivated with 4 µg/ml Mitomycin C (Sigma-Aldrich, M4287) for 2 h, washed three times with PBS, trypsinised (Gibco™, 12605036), and seeded at 20,000 cells/cm². Mitotically inactivated 3T3-J2 were allowed to attach for 5 h and media was changed to a 1:3 ratio of Ham's F-12 Nutrient Mix (Gibco™, 11765054) to DMEM (Gibco™, 41966) with 10% foetal bovine serum (Gibco™, 11563397), 5 µM Y-27632 Dihydrochloride (Bio-Techne Sales Corp, 1254), 25 ng/ml hydrocortisone (Sigma-Aldrich. H0888), 0.125 ng/ml EGF (Thermo Fisher Scientific, 10-605-HNAE50), 5 µg/ml insulin (Sigma-Aldrich, I6634), and 0.1 nM cholera toxin (Sigma-Aldrich, C8052) and human bronchial epithelial cells were seeded at a density of approximately 20,000 cells/cm². To subculture human bronchial epithelial cells, the 3T3-J2 feeder layer was removed with a brief trypsinisation step that does not affect the epithelial cells, followed by three PBS washes. After this initial trypsinisation, a second and longer trypsinisation was carried out to detach the human bronchial epithelial cells, which were subcultured 1:3 on a fresh 3T3-J2 feeder layer. All cells were cultured in a humidified incubator at 37 °C with 5% $CO_2$ and media was refreshed every two days and regularly tested for mycoplasma infection.

### Patient samples
FFPE samples were obtained from the Manchester Cancer Research Centre (MCRC) Biobank, which holds a generic ethics approval (Ref: 18/NW/0092) which can confer this approval to users of banked samples via the MCRC Biobank Access Policy. Samples in this study

were obtained through approval #23_CALO_01. Samples were taken from eight patients between 56 and 74 years old (average age = 65.8). The cohort contained two females, and six males based on patient self-report.

### Genome editing and plasmid construction
To design the sgRNAs for this project, we used the Benchling tool (Benchling.com) that provides sgRNA designs with on-target and off-target scores. We selected batches of four sgRNAs with the lowest off-target score for each gene, prioritising sgRNAs targeting the ORF upstream the first annotated functional domain or within it to ensure complete inactivation of the protein. To discard targeting other loci, we took an approach similar to Drost and colleagues[34] and the top-five predicted off-target loci with the highest off-target scores were sequenced using Sanger sequencing.

The one-step multiplex CRISPR-Cas9 assembly system kit was a gift from Takashi Yamamoto (Addgene Kit #1000000055)[44] was used to construct multiplex CRISPR-Cas9 plasmids used for the generation of cells harbouring knockouts in TP53, CDKN2A, PTEN and KEAP1 genes. gRNAs were assembled into gRNA expression cassettes within pX330 plasmids using the golden gate cloning developed by Sakuma et al.[44]. Firstly, each pair of complementary oligonucleotides (Supplementary Table 2) were annealed by combining 1 µl each of forward and reverse oligonucleotides (100 µM), 1 µl of T4 DNA Ligase Reaction Buffer (NEB, B0202), and 0.5 µl T4 Polynucleotide Kinase (NEB, M0201) in Ultrapure water to a total volume of 10 µl and annealed using a thermocycler (37 °C for 30 min, 95 °C for 5 min, ramp down to 25 °C at 5 °C/minute). Oligonucleotides were then cloned into the relevant pX330S-(2-4) plasmids by combining 0.3 µl of 25 ng/µl plasmid, 0.5 µl of annealed oligonucleotides, 0.2 µl of T4 DNA Ligase Reaction Buffer (NEB, B0202), 0.1 µl Bpil restriction enzyme (Thermo Fisher Scientific, ER1011) and 0.1 µl of Quick Ligase (NEB, M2200) and made to 2 µl with Ultrapure water. Cloning was carried out in a thermal cycler as follows: 3x (37 °C for 5 min, 16 °C for 10 min). The ligation was subsequently used to transform XL1-Blue Competent Cells (Agilent Technologies, 200249) (1 µl ligation product in 20 µl competent cells). pX330S-(2-4) plasmids were then used to clone multiple gRNAs into the gRNA expression cassettes of plasmid pX330A-2, pX330A-3, and pX330A-4 using golden gate cloning. 1.5 µl of each pX330S vector (100 ng/µl) was mixed with 1.5 µl of the relevant pX330A vector (50 ng/µl), 2 µl T4 DNA Ligase Reaction Buffer (NEB, B0202), 1 µl Eco31I enzyme (Thermo Fisher Scientific, ER0291) and 1 µl Quick Ligase (NEB, M2200). Cloning was carried out using a thermocycler: 25x (37 °C 5 min, 16 °C 10 min). An additional digest was included to remove remaining pX330S plasmids from the reaction mix by the addition of 2 µl Buffer G (Thermo Fisher Scientific, BG5) and 1 µl Eco31I enzyme. The digest was incubated at 37 °C for 30 min and inactivated at 80 °C for 5 min. The product was used to transform XL1-Blue Competent Cells.

For a comprehensive list of gRNAs and final pX330 plasmids see Supplementary Table 1. SOX2 overexpression was achieved using the third-generation lentivirus vector, pUltra-hot, which was a gift from Malcom Moore (Addgene #24130). SOX2 cDNA was amplified using restriction site PCR with 5'GAATTC (EcoRI) and 3'GAAGAC (BbsI) flanking sequences and ligated into the multiple cloning site of pUltrahot using a standard restriction digest and ligation, generating the pUltrahot-SOX2 vector.

### Bronchial epithelial cell electroporation and selection
Electroporation of multiplex CRISPR-Cas9 pX330 plasmids into normal bronchial epithelial cells was achieved using the P3 Primary Cell 4D-Nucleofector® X Kit L (Lonza, V4XP-3024) and 4D-Nucleofector™ X Unit (Lonza). 500,000 cells were pelleted and resuspended in 100 µl Nucleofector™ Solution. 5 µg of pX330 plasmid DNA was added and the full volume transferred to a Nucleocuvette™ vessel. Vessels were placed in the 4D-Nucleofector™ X Unit and electroporated using the

 

DC-100 programme, incubated at room temperature for 10 min, and diluted with 500 µl of pre-warmed media before seeding into a T25 containing 4.5 ml of media and an adhered layer of mitotically inactivated mouse embryonic 3T3 fibroblasts, prepared as described in the Cell culture section. Cells were allowed to recover for two days, after which *TP53* mutant selection was achieved by addition of Nutlin-3A (Sigma-Aldrich, SML0580) to a final concentration of 5 nM. Selection was carried out for 5 days, with media changes + Nutlin-3A every 2 days. Mutant bronchial epithelial cell colonies typically appeared 2-3 days following the end of selection.

### Bronchial epithelial cell transduction

The generation of mutant bronchial epithelial cells with stable *SOX2* overexpression was carried out by transduction using lentivirus particles harbouring the packaged pUltrahot-SOX2 vector. Lentivirus was generated by transfecting 90% confluent Lenti-X 293 T cells with 880 ng pUltrahot-SOX2, 572 ng pMDLg/pRRE (Addgene #122251), 308 ng pCMV-VSV-G (Addgene #8454), and 220 ng pRSV-Rev (Addgene #12253) in 2 ml of media and 6 µl of FuGENE HD Transfection Reagent (Promega, #E2311). Virus titres were concentrated using a solution of polyethylene glycol (40% w/v) with NaCl (2.4% w/v) in sterile $H_2O$. The number of virus copies/ml was determined using the Lenti-X™ qRT-PCR Titration Kit (Takara Bioscience, 631235) according to the manual. The same procedure was used to generate lentivirus particles harbouring the packaged pUltrahot vector lacking *SOX2* cDNA which served as an empty vector control. For transductions, 250,000 bronchial epithelial cells were seeded into a T25 containing 5 ml of media and an adhered layer of mitotically inactivated mouse embryonic 3T3 fibroblasts, prepared as described in the Cell culture section. With bronchial epithelial cells still in suspension, a transduction mixture consisting of $4 \times 10^8$ virus copies, 12 mg/ml polybrene, and PBS to 100 µl was added to give a transduction efficiency of >90%. Transductions were carried out for 48 h before fresh media was added, and initial transduction efficiencies were assessed by flow cytometry for mCherry detection.

### PCR genotyping and next generation amplicon sequencing

For genotyping to detect the presence of CRISPR-Cas9 mediated indels, DNA was extracted from mutant bronchial epithelial cells using proteinase K digestion. Approximately 500,000 cells were digested in 500 µl of proteinase K digestion buffer (50 mM KCl, 50 mM Tris-HCl, 2.5 mM EDTA, 0.45% NP40 and 0.45% Tween) with 50 µg Proteinase K (ThermoFisher Scientific, EO0491). Primers for PCR amplification were as follows: CDKN2A_forward 5'-CACCCTGGCTCTGACCATTC-3', CDKN2A_reverse 5'-GCAAGTCCATTTCGGGATTA-3', KEAP1_forward 5'-TACGACTGCGAACAGCGAC-3', KEAP1_reverse 5'-GGCACAGAATCAAAGGTCAC-3', PTEN_forward 5'-TCCAGTGTTTCTTTTAAATACCTGTT-3', PTEN_reverse 5'-GGGGGAGAATAATAATTATGTGAGGT-3', TP53_forward 5'-CAGGAAGCCAAAGGGTGAA-3', and TP53_reverse 5'-CCCATCTACAGTCCCCCTTG-3'. PCR products were sequenced by Sanger sequencing and sequencing traces assessed for the presence of indels. For the assessment of % mutant reads by next generation amplicon sequencing, DNA was extracted using the DNeasy Blood & Tissue Kit (Qiagen, 69504) and target loci amplicons were generating by PCR using the above primers. Amplicons were then pooled to give a single amplicon admixture per mutant cell population. Targeted sequencing was carried out using the MiSeq Nano v2 Kit (Illumina) on the MiSeq desktop sequencer (illumina). Reads were aligned to hg38.

### Western blotting

Whole cell protein lysates were obtained by lysing cell pellets on ice in NP-40 buffer (150 mM NaCl, 1% NP-40, 50 mM tris pH 8) with 10x Protease Inhibitor Cocktail (Sigma-Aldrich, P8340). For sample preparation, 20-40ug of protein was mixed with 4x NuPAGE LDS buffer (Invitrogen, NP0007) and 10x NuPAGE reducing agent (Invitrogen, NP0004) and denatured at 90 °C for 10 min. Equivalent amounts of each sample were run on 4-12% Bis-Tris gels (ThermoFisher Scientific, NP0322), transferred to methanol-activated PVDF membranes and blocked for 1 h at room temperature using 5% Marvel milk, or 5% BSA in TBST buffer. Membrane were immunoblotted with antibodies raised against p53 (SCB, sc-126, Clone DO1, 1:1000), CDKN2A (p16) (CST, 92803, Clone D3W8G, 1:500), KEAP1 (SCB, sc-365626 Clone G2, 1:500), PTEN (SCB, sc-7974, Clone A2B1, 1:500), AKT (CST, 4691, Clone C67E7, 1:1000), pAKT(ser473) (CST, 4060, Clone D9E1:1000), NQO1 (SCB, 32793, Clone A180, 1:1000), SOX2 (Abcam, ab97959, 1:1000) mCherry (CST, 43590, Clone E5D8F, 1:1000), and Vinculin (Sigma, V9264, Clone hVIN-1, 1:10,000). Secondary antibodies were goat anti-rabbit IgG HRP (Agilent Technologies, P0440801-2), or rabbit anti-mouse IgG HRP (Agilent technologies, P044701-2) at 1:5000. Blots were developed using SuperSignal™ West Pico PLUS Chemiluminescent Substrate (ThermoFisher Scientific, 34577).

### Proliferation assays

Crystal violet cell proliferation assays were carried out in 6-well tissue culture plates. 50,000 bronchial epithelial cells were seeded into one well of a 12-well tissue culture plate containing an adhered layer of mitotically inactivated mouse embryonic 3T3 fibroblasts, prepared as described in the Cell culture section and grown in a humidified incubator at 37 °C with 5% CO2. Plates were harvest, fixed and stained at 24-hour intervals for 3 days. Briefly, 3T3-J2 cells were first removed with a quick trypsinisation, leaving bronchial epithelial cell colonies attached. Colonies were washed twice with ice-cold PSB and fixed for 10 min with ice-cold methanol. Colonies were then stained with 0.1% crystal violet dissolved in 20% methanol (w/w) in $H_2O$ for 15 min, de-stained under running water and dried overnight. Plates were imaged using an Epsom flatbed scanner and images analysed using Fiji (ImageJ) to give an output of percent confluency per well.

### Soft-agar colony forming assays

Colony forming assays were used to investigate anchorage-independent growth. Base agar was generated by mixing a 1:1 ratio of 1% agar (w/v) in sterile water with cell culture media consisting of equal parts 2x DMEM (Gibco, 12800017) and 2x DMEM/F12 (Gibco, 12400024) supplemented with 20% FBS, 1uM Y-27632 (Bio-Techne, 1254/10), 0.5 ng/ml EGF (Gibco, PHG0311), 1ug/ml hydrocortisone (Sigma-Aldrich, H0888-1G), 10ug/ml insulin (SLS, I6634), 0.2 nM cholera toxin (Sigma-Aldrich, C8052), and 10% penicilin-steptomycin (Gibco, 15140122). Top agar was generated by mixing a 1:1 ratio of 0.7% agar (w/v) with cell culture media consisting of equal parts 2x DMEM (Gibco, 12800017) and 2x DMEM/F12 (Gibco, 12400024) with 10% sodium bicarbonate solution (v/v) (Gibco, 25080094). 1 ml of base agar was added each well of a 6-well tissue culture plate and set for 5 min. A single cell suspension of 50,000 epithelial cells was mixed with 1 ml top agar and seeded on top of the set base agar. Plates were incubated in a tissue culture incubator at 37 °C with 5% $CO_2$ for 4 weeks and fed with 300 µl complete epithelial cell culture media once weekly. Colonies were visualised by staining with 0.005% crystal violet solution for 2-hours and 20x brightfield images of each well were captured using an inverted microscope. Colony quantification was carried out using Fiji where crystal violet-stained foci with a circumference >50 µM were counted as a colony.

### Real-time Caspase-3/7 activity assays

Caspase-3/7 activity assays were carried out to determine the apoptotic effect of viral transduction using the pUltrahot-*SOX2* vector on mutant human bronchial epithelial cells. To capture apoptosis immediately after virus transduction, 3200 bronchial epithelial cells were seeded into one well of a 96-well tissue culture plate containing an adhered layer of mitotically inactivated mouse embryonic 3T3 fibroblasts, prepared as described in the Cell culture section with 50 µl of

media. With epithelial cells still in suspension, a transduction mixture consisting of $4 \times 10^6$ virus copies, 12 mg/ml polybrene, 5 μM NucView 488 caspase-3 substrate (Biotium, 30029) and media to 50 μl was added. Reversine (Sigma-Aldrich, R3904) was added to a final concentration of 2 nM in positive control wells to induce caspase activity. Wells with transduction mixture containing equal amounts of the pUltrahot empty vector were used to control for background levels of virus mediated apoptosis. Plates were incubated in the Incucyte Live-Cell imager for 48 h, with images captured every 60 min.

## Collagen-I invasion assays

Invasion assays were performed using human bronchial epithelial cells grown on collagen discs containing primary human pulmonary fibroblasts (Lonza Bioscience, ID: HLF-2, batch:20TL356516). Disc generation and contraction was carried out the method published by Timpson et al.[15]. Following this method, for 12 discs, $1 \times 10^6$ pulmonary fibroblasts were resuspended in 3 ml FBS (Fisher Scientific, 16-107-078) and the whole volume added to a mixture of 20 ml collagen-I (5 mg/ml) (Enzo Life Sciences, ALX-522-435-0100), 5 ml sterile $H_2O$, 3 ml 10x MEM (Gibco, 11430030), and 0.22 M NaOH. 0.22 M NaOH was added to increase the pH of the collagen mixture to achieve a transition from yellow to salmon pink (approximately pH 7). All components were prepared in a sterile cell culture hood and always kept on ice. 2.5 ml of fibroblast-collagen mixture was added per $3mm^3$ cell culture dish and allowed to polymerise for 20 min in a 37 °C humidified cell culture incubator. Following this, 1 ml of prewarmed DMEM media (Gibco, 41966029) supplemented with 10% FBS, 5% penicillin-streptomycin (Gibco, 15070063) and 5% GlutaMAX (Gibco, 35050061) was added. The next day, an additional 1 ml of media was added to each dish and media was changed every 2 days until the discs contracted to the size of one well of a 24-well tissue culture plate (approximately 8 days). Following contraction, collagen discs were transferred to a 24-well dish and 40,000 epithelial cells were seeded onto each disc in 1 ml of PneumaCult™-Ex Plus expansion media (STEMCELL Technologies, 05040). Media was changed every 2 days for 1 week, after which each disc was suspended on a PET membrane in one well of a 6-well plate using tissue culture insert (Sarstedt, 83.3930). 2 ml of complete PneumaCult™-ALI differentiation media (STEMCELL Technologies, 05001) was then added to the basal compartment of each well to generate an air-liquid interface, and media was changed every 2 days. Collagen-I invasion assays were cultured at 37 °C in a humidified incubator with 5% $CO_2$ for 3 weeks.

## Air-liquid interface culture and processing

Air liquid interface organotypic cultures were generated using human bronchial epithelial cells. Cultures were grown on Falcon® Permeable Supports with PET membranes (pore size 0.4 μm) (Corning, 353095). Firstly, PET membranes were coated with 20 μg type-I bovine collagen. Briefly, 3 mg/ml PureCol® solution (CellSystems, 5005) was diluted 1:30 in sterile PBS and 200 μl added to each PET membrane and incubated for 1 h at room temperature. Collagen solution was removed, and membranes were washed gently with PBS. 30,000 bronchial epithelial cells were pelleted and resuspended in 100 μl Airway Epithelial Cell Growth Medium (PromoCell, C-21160), after which the whole amount was seeded onto a collagen coated Falcon® Permeable Support placed within one well of a 24-wel plate. and 500 μl of PneumoCult™-ALI complete Basal Medium (STEMCELL Technologies, 05100) was added to the basal compartment. Epithelial cells were expanded for 7 days under submerged conditions with media changes every 2 days. On day 7, apical media was removed, and basal media replaced with 500 μl of PneumoCult™-ALI Complete Maintenance Medium (STEMCELL Technologies, 05100). Cultures were allowed to differentiate for 21 days with basal compartment media changes every 2 days. Once complete, cultures were processed for downstream analysis. For histological assessment, cultures were fixed for 30 min in

4% paraformaldehyde, embedded in agarose, and processed into paraffin blocks for sectioning. For gene expression analysis, cultures were washed in PBS and RNA was extracted using the RNeasy Mini Kit (Qiagen, 74104) according to the manufacturer's instructions.

## Cisplatin sensitivity experiments

On day 21 after removal of apical media from ALI cultures, cisplatin (Sigma, C-2210000) dissolved in 0.9% sodium chloride was added the specified concentrations to the basal media.

To monitor LDH activity in the basal medium, 10 μl of basal media were removed at 24, 48 and 72 h of cisplatin treatment and diluted in 100 μl LDH storage buffer (200 mM Tris-HCl, pH 7.3, 1% BSA, 10% Glycerol) and stored at -20°C for later LDH activity quantification. To normalise LDH activity in basal media, ALI cultures were lysed with 0.2% Triton-X100 in ALI culture media for 30 min under constant shaking. 10 μl of the resulting lysate were diluted in 100 μl of LDH storage buffer. LDH activity in the basal media and ALI culture lysates was determined with the LDH-Glo™ assay (Promega) according to the manufacturer's instructions in an opaque bottom white 96-well plate compatible with chemiluminescence. Normalised LDH activity was calculated as the ratio of LDH activity in the volume of basal media used in ALI cultures (500 μl) and the total LDH activity in the ALI lysate for each time-point and genotype.

Cell viability and caspase 3/7 activity were determined using the CellTiter-Blue® and Apo-ONE® assays (both from Promega). Briefly, 20 μl of CellTiter-Blue® were added to 100 μl of ALI differentiation media in the apical compartment of the insert and incubated at 37 °C for one hour in a humidified tissue culture incubator. Fluorescence was measured by transferring the resulting lysate to a clear bottom black 96-well plate at 560 nm(ex)/590 nm(em). After removing the CellTiter-Blue® reagent, 100 μl of the Apo-ONE® reagent were added to the apical compartment according to manufacturer's instructions and incubated for 2 h at room temperature under constant shaking. Apo-ONE fluorescence was measured by transferring the resulting lysate to a clear bottom black 96-well plate 485 nm(ex)/ 520 nm(em).

## Immunofluorescence and immunohistochemistry

4 μm FFPE sections were cut from FFPE blocks from ALI organotypic cultures, collagen-I invasion assays and clinical patient samples. Immunofluorescence staining was used for anti-MUC5AC (Invitrogen, MA5-12178, Clone 45M1, 1:400) anti-acetylated tubulin (Sigma, T6793, Clone 611B1, 1:400), anti-vimentin (Invitrogen, MA5-11883, Clone V9, 1:250) and anti-EPCAM (Abcam, ab223582, Clone EPR20532-225) (1:500). Antigen retrieval was performed in a pressurised vessel for 20 min at 95 °C using Target Retrieval Solution, pH 9 (Agilent, S2367). 400 μl/slide of primary antibody was incubated on slides overnight at 4 °C. Secondary antibodies, including goat anti-mouse IgG1 Alexa Fluor 488 (Invitrogen, A-21121), goat anti-rabbit IgG Alexa Fluor 647 (Invitrogen, A32733), and goat anti-mouse IgG2b Alexa Fluor 647 (Invitrogen, A-21242), were diluted to 1:400 in PBS with 5% BSA and incubated on slides for 1 h.

Immunohistochemistry was used for anti-TTF1 (Abcam, ab76013, Clone EP1584Y, 1:100), anti-cytokeratin 5/6 (Invitrogen, MA191106, Clone D5/16B4, 1:100) and anti-p63 (Abcam, ab124762, Clone EPR5701, 1:400) antibodies, anti-CC10 (SCB, 365992, Clone E11, 1:2000), anti-involucrin (Invitrogen, MA5-11803, Clone SY5, 1:200), anti-SOX2 (Abcam, ab92494, Clone EPR3131, 1:100), and anti-mCherry (Novus, Clone NBP2-25157, 1:500). Antigen retrieval (ER 1, 20 min) and staining was performed using the Leica Bond RX automated platform using the classical IHC-F protocol with antigen retrieval achieved using BOND™ Epitope Retrieval solution 1 (Lecia, AR9961) for 20 min. CC10 and involucrin staining protocols included an additional 30-minute blocking step with casein (Sigma-Aldrich, B6429) and 5% bovine serum, respectively. Chromogenic 3,3-diaminobenzidine (DAB)

staining was achieved using the BOND™ Polymer Refine Detection Kit (Lecia, DS9800).

## Image acquisition and analysis

All slides were scanned at 20x using the Olympus VS120 (fluorescence), or Olympus VS200 (brightfield). The images were analysed using HALO image analysis software (Indica Labs), with the Multiplex IHC 2.0 module. Percent positive MUC5AC, CC10, and p63 cells were calculated by dividing the number of positive cells by the total number of nuclei per analysis region. Percent acetylated tubulin coverage was calculated by dividing the length of acetylated tubulin positive epithelium by the total length of epithelium analysed. A fraction of mCherry positive epithelium was calculated manually by dividing the length of mCherry positive epithelium by the total length of the epithelium. For invasion assays, measurements and manual counting of single invading epithelial cells (EPCAM$^+$VIM$^-$, or EPCAM$^+$VIM$^+$ cells), was achieved using the OlyVIA slide viewing software (Olympus). To calculate the number of invading cells per mm$^2$, the number of invading cells was divided by the length of collagen disc cross-section. To avoid counting cells still attached to the main epithelial surface, EPCAM$^+$VIM$^-$, or EPCAM$^+$VIM$^+$ cells within 100 μm of the epithelium base were discounted.

## Gene expression analysis by qPCR

qPCR was carried out using the Power SYBR Green PCR chemistry (Thermo Fisher Scientific, 4367659). cDNA was from total RNA using Superscript III (Thermo Fisher Scientific, 18080093) according to manufacturer's instructions. The following oligonucleotides were used: SPRR2A (forward: gtatccaccgaagagcaagtaa, reverse: ggaacgaggtgagccaaata) and SPRR3 (forward: agcagaagaccaagcagaag, gacacagaaaacagatgggaaga) and beta-actin (forward: tggatcagcaagcaggagtatg, reverse: gcatttgcggtggacgat).

## RNA-sequencing

RNA was extracted using the RNeasy Mini kit (Qiagen, 74104) according to the manufacturer's instructions. Library preparation and RNA sequencing was carried out by the CRUK Manchester Institute Molecular Biology Core Facility. A RIN value of >7 were selected for sequencing. Library preparation was performed using 100 ng of total RNA using the Lexogen QuantSeq 3′ mRNA-Seq Library Prep Kit. Single-end sequencing with 100 bp read length was performed using a Novaseq 6000 sequencer (Illumina). Basecalls were converted to fastq files using bcl2fastq (Illumina). Fastq files were trimmed for adapter sequences using the autodetect feature in trim_galore (version 0.6.5) and aligned to GRCh38 using Star aligner (version 2.5.1b). BAM alignments were quantified in R (version 3.6.1) using featureCounts from the Rsubread library (version 2.0.1). PCA plots were produced using regularised log-transformed read counts from DESeq2 (version 1.26). Two samples from donor 1 were identified as outliers and removed from the analysis (1 from wild type and 1 from TC + P). Due to the sequencing set-up, it was not possible to perform OLS regression for batch effect removal on the collected data. Surrogate variable analysis (SVA) was employed to identify and account for confounding covariates within the combined dataset. DESeq2 was used to perform comparisons between conditions within the combined dataset, including the SVA covariates in the model.

## Weighted gene co-expression network analysis

Data used for WGCNA were pre-processed from combined raw counts by filtering out low-count genes (>10 reads allocated in 90% of samples). Raw counts were subsequently normalised using the vst function (DESeq2)[45] and surrogate variable effects removed using the removeBatchEffects function (limma)[46]. To filter for only those genes differing between genotype, a mixed effects model (gene ~ genotype), with donor info stated as a random effect, was compared to a null model (gene ~ 1) using ANOVA; genes passing an adjusted p-value filter (pAdj <0.05) were selected for further analysis. WGCNA (version 1.72-1)[18] was performed using dynamic power estimate optimisation (powers tested: 2-30). Gene-module allocations using the full dataset were used for further analyses (genes not allocated to a module reside within the "grey" module). Optimised power estimates were used for cross-validation with the sampledBlockwiseModules function (100 iterations of 80% samples across either 2000 or 80% of the input genes, depending on which was lower). Module stability/preservation statistics were calculated for the cross-validation results using the modulePreservation function. To assess heterogeneity between donors, WGCNA and cross-validation was also performed for each individual donor within each subset using a consistent power estimate for each donor. Gene expression within each module was plotted as heatmaps using scaled-normalised counts with the ComplexHeatmap package (version 1.72.1)[47]. Cross-module comparisons were plotted using module eigengenes. Overenrichment analysis (ORA) of the module genes was performed using the clusterProfiler package (version 4.6.2)[48] with gene sets from Hallmark signatures from MSigDB.

## Gene set enrichment analysis, overrepresentation analyses and publicly available LUSC sample databases

Preranked GSEA results were calculated for pairwise comparisons by fgsea (version 1.24.0) using the results defined by DESeq2, subsetting of genes by pAdj<0.05 and ranking using Log2FoldChange. Condensation of GO terms was performed using simplifyEnrichment[49].

Differentially expressed genes in LUSC patient samples with OSR-targeting alterations from the LUSC CPTAC[8] and TCGA[9] cohorts were downloaded via cBioportal[50] and enrichment analysis carried out using the shinyGO application[51].

Preinvasive lesion gene-expression data from Mascaux et al.[22] (GSE33479) was downloaded using the XTABLE application[23].

## Correlation and overlap analyses between LUSC samples and mutant hBECs

Here we downloaded the CPTAC-LUSC cohort from GDAC. The supplied gene counts (TPM unstranded) for each patient were constructed into a count file. Patient metadata were obtained from the original publication supplemental information. Mean counts for each subtype were obtained by removing all "mixed subtype" samples and taking an average across samples within a single subtype for each gene.

To assess the simultaneous influence of the three pathways (SOX2, KEAP1 and PTEN), we extracted the significantly enriched terms in the TC + PKS vs TC, assessed via preranked GSEA using the statistics from independent DESeq2 comparisons. The extracted terms (GO Biological Process) were then used as the basis for ssGSEA, using the mean counts from the CPTAC-LUSC cohort and mean counts from the ALI. Correlation of ssGSEA results between LUSC and ALI samples was measured using Pearson's correlation, p-value adjustment performed using the Benjamini-Hochberg method.

To compare the overlap between the GO terms (Biological Process) regulated by each pathway individually, we compared the significantly regulated terms in the TC + S vs TC, TC + K vs TC and TC + P vs TC comparisons and the significantly enriched terms between obtained from the TCGA cohort and extracted using the differentially expressed genes in samples with and without SOX2 amplification, samples with and without alterations in the OSR pathway (NFE2L2, KEAP1 and CUL3), and samples with and without PTEN alterations. Chi-square test was carried out to calculate the statistical significance of the overlap, using all the GO terms with at least one gene mapped as the background list.

## Quantification and statistics

Statistical testing of cell biology data including proliferation, colony forming, invasion assays, RT-qPCR, and the quantification of clinical

sample and organoid immunostaining was carried out using the GraphPad Prism 9 software. Statistical comparisons were performed using a one-way ANOVA with *post hoc* tests for multiple comparison correction. In some cases, specific pairwise comparisons were selected from the larger dataset to increase power by reducing the number of comparisons made. These selections were justified in that they removed biologically meaningless, or redundant comparisons within the dataset. Significance was determined by p value < 0.05. Data was presented as means with standard deviation. n denotes the number of biological replicates.

All statistical testing was two-sided, analysis of normality of distributions was tested using the Kolmogorov-Smirnov test, and replicate measurements were taken from distinct samples.

### Reporting summary

Further information on research design is available in the Nature Portfolio Reporting Summary linked to this article.

### Data availability

The ALI culture raw RNA-seq data generated in this study are available with unrestricted access in the BioProject database under accession number:PRJNA1043668 [https://www.ncbi.nlm.nih.gov/bioproject/?term=PRJNA1043668]. The datasets that have been previously published and used in this article include. GSE33479 (Premalignant lesion RNA microarray data). PDC000234 (CPTAC proteomics data). TCGA-LUSC (TCGA RNA-seq data). Source data are provided with this paper.

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

## Acknowledgements

This project has been funded by the National Centre for the Replacement, Refinement & Reduction of Animals in Research (NC/W001284/1 to C.L.G), the Rosetrees Trust (M767 to C.L.G.), the Cancer Research UK Lung Cancer Centre of Excellence (A25146 to C.D.) and the Cancer Research UK Manchester Institute (C5759/A20971 to C.D.). We would like to thank the Molecular Biology, Histopathology, Flow Cytometry and Scientific Computing core facilities at the Cancer Research UK Manchester Institute, the National Biomarker Centre, and the Manchester Cancer Research Centre Biobank.

## Author contributions

J.O. contributed to conceptualization, data curation, formal analysis, investigation, methodology, validation, visualization and writing-original draft, review and editing. R.S. contributed to conceptualization, data curation, formal analysis, methodology, software, validation. S.S. contributed to conceptualization, data curation, formal analysis, methodology, software, supervision and validation. A.O. contributed to data curation, investigation, methodology, project administration, resources, validation and writing- review and editing. A.C. contributed to data curation, investigation and resources. C.D. contributed to conceptualization, funding acquisition, methodology, resources, supervision and writing-review and editing. C.L.G. contributed to conceptualization, data curation, formal analysis, funding acquisition, methodology, project administration, resources, supervision, validation, visualization and writing-original draft, review and editing.

## Competing interests

The authors declare no competing interests.
