## [Peer Review File · Nature Communications]

A human model to deconvolve genotype-phenotype causations in lung squamous cell carcinoma

Corresponding Author: Dr Carlos Lopez-Garcia

Version 0:

Reviewer comments:

Reviewer #1

(Remarks to the Author)

This is an interesting methodological paper aimed at developing an in vitro model for human lung squamous cell carcinoma (LUSC). The authors used a very limited number of bronchial epithelial cells (hBECs) just from three human donors. They carried out an extensive combination of different gene modifications and tested the behavior in vitro of the resulting cell lines. Then they went ahead and did an extensive bulk RNAseq analysis to explore potential interactions among the different modifiers. The results of these analyses were mostly confirmatory, and the new information was just incremental. This can be the initial step for a potentially useful system to study tumorigenesis. Since there is a general lack of new scientific or medical information in these results, this manuscript is not suitable for Nature Communications and it seems more appropriate as a methodological paper. In any case, the study appears underdeveloped, too preliminary, and, consequently, of limited value since: (1) it lacks human validations from actual LUSC samples; (2) it is unclear how just three donors-derived cell lines could account for the tremendous heterogeneity of human cancer; (3) the complete lack of any attempt to use this model system as a base for a more complex scenario akin to the microenvironment of the tumor.

Reviewer #2

(Remarks to the Author)

In this study Ogden et al. describe an in vitro model system for lung squamous cell carcinomas (LUSC). Human bronchial epithelial cells (hBECs) were isolated from three non-smoker healthy individuals and manipulations of genes typically found in human LUSC were performed; these include ubiquitous loss of TP53 and CDKN2A, and combinations of SOX2 overexpression as well as PTEN and KEAP1 depletions.

The authors were able to cultivate these cells, which upon gene manipulations showed an aggressive profile, resembled cancer cells and the histology of LUSC. SOX2 expression was found to induce early preinvasive LUSC stages, and cooperative gene manipulations showed classical LUSC histology. The authors performed transcriptional studies and confirm signaling pathways associated with these gene alterations in human LUSC. The transcriptional and cellular profiles were found to be similar in three distinct hBECs used for this study

LUSC is a very frequent lung cancer subtype, and targetable treatments have not been identified. It is therefore of high relevance to generate adequate model systems to study this deadly disease. The authors present a very elegant in vitro system, that seems to allow for in-depth functional in vitro studies. The authors describe the technical set-up very well. Some aspects could be addressed as part of this work, to further corroborate the validity of this model system:

1. The authors could refer to other in vitro approaches to generate squamous cell cancer system and discuss advantages of their system (e.g., Kawai et al., Scientific Reports, 2021; Pan et al., Cancer Cell, 2022).
2. Transcriptional profiles: the authors studied the transcriptional profiles of their model systems and point to pathways that are known to be associated with the respective gene modulations (SOX2 expression, KEAP1 and PTEN depletion). It would be important to perform comparative studies with data from patients with LUSC (available as part of the TCGA or ICGC) and to subset for specific genomic alterations in patients (SOX2 expression/amplification, KEAP1 and PTEN alterations) and provide a side-to-side comparison with the respective in vitro model system. A general comparison with human LUSC patient data would also help to further confirm that their model system truly represents the classical profile of LUSC.
3. Establishing an in vitro model system could facilitate testing the cells for drug sensitivity. It would be relevant to show if this model systems could allow for drug screens and functional readouts. The authors could address how these different

gene alterations may impact response to drugs (e.g. chemotherapy).

Reviewer #3

(Remarks to the Author)

I am grateful to the Editorial team of Nature Communications for granting me and my junior colleague the opportunity to conduct a peer review of the manuscript authored by Ogden et al. The study successfully developed a new in-vitro human model for investigating the intrinsic phenotypes of genetically dysregulated pathways in LUSC. In accordance with the authors' expectations, I am of the opinion that this model has the potential to provide a robust means of understanding the complex relationships between genotype and phenotype. Furthermore, it can facilitate the development of target therapies and diagnostic strategies for LUSC.

I would like to discuss several comments:

1. Many investigators have lately employed organoids for basic cancer research. It would be beneficial if the author could elucidate the pros and cons of this model in contrast to organoid in the discussion section.
2. What is the authors' opinion on the applicability of this concept to cancer patients? For instance, can you induce normal cells to differentiate in a manner that replicates the specific characteristics of the cancer present in a specific patient?
3. I concur with your viewpoint regarding the utilization of models for basic research. How do you foresee the implementation of this concept in clinical practice? What specific tests do you propose for acquiring hBECs from patients to build this model? If you successfully model cancer using hBECs from a particular patient, how do you envision its application in terms of diagnosis and treatment? Please engage in a discussion.
4. The authors employed a genetic engineering technique, utilizing CRISPR-Cas9 for genetic modification in this model. How did the authors take measures to prevent the influence of off-target effects on their results? What measures were implemented to ensure the specificity of gene editing?
5. The study discovered that overexpression of SOX2 inhibits proliferation of cells in 2D cultures, but enhanced anchorage-independent growth and invasiveness. How do the authors reconcile this paradox, and what further experiments might clarify the role of SOX2?
6. The results demonstrate heterogeneity among donors, especially in terms of colony formation and the expansion of p63+ cells. How do the authors interpret and clarify the variation between donors, and what are implications for the generalizability of the findings?
7. The study focuses mainly on the SD, PI3K/Akt, and OSR pathways. Are there any further pathways or genetic alterations that could have a significant impact on LUSC but were not taken into account in this model? Do authors have precise criteria for selecting three pathways for the creation of this model?
8. LUSC typically originates from cells that have been exposed to chronic inflammation, such as in heavy smokers or individuals with chronic obstructive pulmonary disease (COPD). I am curious about the potential alterations in the outcomes if this model were to be implemented on hBEC derived from those specific patients. What are the authors' opinions regarding this matter? Is there any likelihood that the LUSC model derived from heavy smoker donors exhibits similarity to the LUSC of actual patients?
9. The authors stated that the amplification of SOX2 is linked to the remodeling of the tumor microenvironment. Nevertheless, this particular model lacks TME, and the authors drew this conclusion based on the downregulation of CIITA. I fully disagree with this interpretation.
10. I believe that the results of WGCNA are not the main findings of this study. What if the authors relocate the results as additional or supplementary data?

Reviewer #4

(Remarks to the Author)

Version 1:

Reviewer comments:

Reviewer #1

(Remarks to the Author)

The revisions have substantially improved the manuscript and addressed several of the key concerns raised by this reviewer. The inclusion of additional data validating the human relevance of the model using analyses of TCGA and CPTAC datasets has strengthened the validity of the developed model. Although some limitations remain, especially those related to capturing heterogeneity and the impact of the tumor microenvironment, overall, the revisions have enhanced the clarity and quality of the paper in providing an additional model system to study LUSC. Therefore, I recommend publication in Nature Communications.

Reviewer #2

(Remarks to the Author)

Ogden et al. revised their manuscript addressing most of the comments suggested by the reviewers. Overall, the authors provide a good amount of additional data. I have no further comments.

Reviewer #3

(Remarks to the Author)

I carefully reviewed the updated version and determined that all the weaknesses I stated were properly addressed. While the model's applicability in clinical settings remains uncertain, it appears to hold potential for basic research.

Reviewer #4

(Remarks to the Author)

REBUTTAL LETTER (Manuscript NCOMMS-24-11773-T)

Reviewer #1 (Remarks to the Author): Expert in air liquid interface organotypic cultures and cancer development

This is an interesting methodological paper aimed at developing an in vitro model for human lung squamous cell carcinoma (LUSC). The authors used a very limited number of bronchial epithelial cells (hBECs) just from three human donors. They carried out an extensive combination of different gene modifications and tested the behavior in vitro of the resulting cell lines. Then they went ahead and did an extensive bulk RNAseq analysis to explore potential interactions among the different modifiers. The results of these analyses were mostly confirmatory, and the new information was just incremental. This can be the initial step for a potentially useful system to study tumorigenesis. Since there is a general lack of new scientific or medical information in these results, this manuscript is not suitable for Nature Communications and it seems more appropriate as a methodological paper. In any case, the study appears underdeveloped, too preliminary, and, consequently, of limited value since: (1) it lacks human validations from actual LUSC samples; (2) it is unclear how just three donors-derived cell lines could account for the tremendous heterogeneity of human cancer; (3) the complete lack of any attempt to use this model system as a base for a more complex scenario akin to the microenvironment of the tumor.

Response: We are sorry that the reviewer thinks the manuscript is unsuitable for Nature Communications and of limited value. We want to address such views based on the three reasons provided by the reviewer.

The reviewer argues that the manuscript lacks validation from human samples. Throughout the manuscript, we have compared our data with LUSC samples. Namely, we compared the expansion of p63+ve cells in our ALI cultures with premalignant lesions, and we frequently compared our gene-expression data with the TCGA and CPTAC databases in throughout the manuscript. For instance, to validate SOX2 correlation with MHC-II expression or downregulation of interferon responses in LUSC with alterations in Nrf-2 pathway components. Importantly, we also compare our hypothesised evolutionary trajectory with premalignant lesions from patients in figure 8.

However, we agree that we have not validated our results globally to support our central claim that our model recapitulates the classical LUSC subtype. To address this, we have conducted a global transcriptomic analysis comparing the gene-expression data in our mutants with the CPTAC cohort. We found a significant positive correlation between the transcriptomes of our TC+PKS mutants (the mutant that recapitulates invasive carcinomas) and the classical subtype. Interestingly, we also found a positive correlation between TC+PKS mutants with the LUSC EMT subtype. Additionally, we have carried out a side-by-side comparison between the gene sets regulated by each pathway individually in our mutant hBECs and LUSC patient samples with and without alterations in the same pathways. This data can be found in the Supplementary Figure 8.

The reviewer argues that it is unclear how three donors account for the tremendous heterogeneity of human cancer. We might have misled the reviewer when with our sentences:

“To capture some inter-person variation and avoid smoking-induced pre-existing alterations, mutants were generated using hBECs from three never-smoking donors without reported airways disease”

“Mouse models are valuable for understanding basic LUSC biology. However, availability of resources and inter-species differences hinder the modelling of increasingly complex genotypes, evolutionary trajectories and inter-patient heterogeneity.”

These sentences seem to wrongly indicate that we set out to model inter-patient heterogeneity from the beginning. We are not claiming that by using three donors, we accounted for the heterogeneity of cancer. None of the existing genetic cancer models, whether lung or otherwise, capture that tremendous heterogeneity. Instead, we are claiming that the versatility of our model can be leveraged to investigate the origins of such heterogeneity, for instance, the molecular subtypes. By using three donors, we intended to provide an initial assessment of the influence of the donor in the phenotypes we analysed, which appears to be limited. In this regard, our model does not differ from previous approaches to model cancer from human primary non-immortalised cells of origin. Three is the maximum number of donors used in similar approaches to model colon cancer (PMID: 25706875, PMID: 25924068), melanoma (PMID: 35482859) and medulloblastoma (PMID: 31996670).

Additionally, we have modified the sentences mentioned previously to enhance accuracy as follows:

*“To capture **major phenotypic differences caused** inter-person variation....”*

*“Mouse models are valuable for understanding basic LUSC biology. However, availability of resources and inter-species differences hinder the modelling of increasingly complex genotypes, evolutionary trajectories and **molecular subtypes**”*

The reviewer is sceptical about the complete lack of any attempt to use the system to investigate the effect of the tumour microenvironment. In fact, due to the important influence of stromal components on the process of invasion, we used the collagen-disc assay, which incorporates pulmonary fibroblasts to investigate invasiveness. However, this might not be clear in the main text, and we have modified it accordingly.

Investigating the effect of the microenvironment is precisely the next step of our project. Furthermore, we believe that this system provides a unique opportunity to build a model progressively and dissect how different microenvironment components cooperate in LUSC development. However, we think that undertaking this complex research would have been a separate project itself. We intended to develop a model that recapitulates the epithelial architecture and can be progressively upgraded to incorporate the complex heterotypic interactions within the tumour microenvironment, in a similar manner to the report by Nikolaev and colleagues (Nature 2020, PMID:32939089), in which the authors report the mini-colon system that does not incorporate stromal components. This report and ours, although with a different scope, intend to report and validate new systems with enormous potential that broaden the range of research possibilities, including the study of tumour-stromal interaction.

Reviewer #2 (Remarks to the Author): Expert in lung cancer genomics, evolution, and development

In this study Ogden et al. describe an in vitro model system for lung squamous cell carcinomas

(LUSC). Human bronchial epithelial cells (hBECs) were isolated from three non-smoker healthy individuals and manipulations of genes typically found in human LUSC were performed; these include ubiquitous loss of TP53 and CDKN2A, and combinations of SOX2 overexpression as well as PTEN and KEAP1 depletions.

The authors were able to cultivate these cells, which upon gene manipulations showed an aggressive profile, resembled cancer cells and the histology of LUSC. SOX2 expression was found to induce early preinvasive LUSC stages, and cooperative gene manipulations showed classical LUSC histology. The authors performed transcriptional studies and confirm signaling pathways associated with these gene alterations in human LUSC. The transcriptional and cellular profiles were found to be similar in three distinct hBECs used for this study

LUSC is a very frequent lung cancer subtype, and targetable treatments have not been identified. It is therefore of high relevance to generate adequate model systems to study this deadly disease. The authors present a very elegant *in vitro* system, that seems to allow for in-depth functional *in vitro* studies. The authors describe the technical set-up very well. Some aspects could be addressed as part of this work, to further corroborate the validity of this model system:

1. The authors could refer to other *in vitro* approaches to generate squamous cell cancer system and discuss advantages of their system (e.g., Kawai et al., Scientific Reports, 2021; Pan et al., Cancer Cell, 2022).

Response: We thank the reviewer for raising those reports, which are now mentioned in our discussion. The scope of the article by Kawai and colleagues (Scientific Reports, 2021, PMID:34934075) is substantially different from ours. They developed a system to optimise the conservation of keratinising structures in LUSC organoids generated from cell lines. However, the authors did not consider the effect of tumour genetics on cancer organoid morphology, an undertaking that was beyond their scope and is much more feasible using an isogenic reverse-genetics approach such as the one presented in our report.

On the other hand, the scope of the report by Pan and colleagues (Cancer Cell, 2022, PMID:36525973) aligns better with our objectives, as it aims to obtain the genotype-phenotype causation of a specific LUSC driver gene, *KMT2D*. However, unlike our system, Pan and colleagues implemented an *in vivo* approach, and the *in vitro* work they carried out was limited. Nevertheless, the *in vitro* experiments they presented (such as the sensitivity studies to SHP2 and pERRB inhibition) correlated well with the *in vivo* results and anticipate that our human system could have been used to address the function of *KMT2D* in LUSC. Aside from the *in vivo* vs *in vitro* differences, our system has the advantage of being human and has a versatility that enables us to model the effect of LUSC carcinogenesis at a higher genetic resolution. This advantage will facilitate enormously the identification of genes that drive non-classical LUSC subtypes.

More specifically, Pan and colleagues found that both *Kmt2d* and *Pten* inactivation in a *Trp53* knock-out background drive tumorigenesis after a long latency period and that *Kmt2d*-driven tumours are more sensitive to RTK-RAS pathway inhibition. However, the authors did not investigate whether genomic events potentially acquired during the latency period cooperate with both genes to fuel tumour progression. For instance, are *Pten*- and *Kmt2d*-driven tumours independent of *Sox2* amplification, or activation of the Nrf-2 pathway? In addition, the authors did not map the phenotypic landmarks of the tumours to molecular or histological LUSC subtypes. For instance, do *Pten*- and *Kmt2d*-driven tumours drive different molecular subtypes?

Addressing the former questions was probably not part of the remit of this remarkable manuscript. However, our novel human model constitutes a more versatile system to identify genes that cooperate with *Kmt2d* in LUSC progression.

2. Transcriptional profiles: the authors studied the transcriptional profiles of their model systems and point to pathways that are known to be associated with the respective gene modulations (SOX2 expression, KEAP1 and PTEN depletion). It would be important to perform comparative studies with data from patients with LUSC (available as part of the TCGA or ICGC) and to subset for specific genomic alterations in patients (SOX2 expression/amplification, KEAP1 and PTEN alterations) and provide a side-to-side comparison with the respective in vitro model system. A general comparison with human LUSC patient data would also help to further confirm that their model system truly represents the classical profile of LUSC.

Response: We have added new data in Supplementary Figure 8 which shows overall correlations between the transcriptomes of LUSC subtypes and the mutants. To do this, we have taken all the gene sets shown in Figures 5 and 6 (GO Biological Processes regulated by SOX2 overexpression, *PTEN* knock-out, and *KEAP1* knock-out simultaneously). Next, we calculated the correlations between the enrichment scores of the same gene sets in each LUSC subtype (PMID:34358469) and in our organotypic cultures generated with mutant hBECs. We expected a significant positive correlation between the LUSC classical subtype and the TC+PKS mutant, as this is the most aggressive mutant and the CPTAC cohort comprises invasive LUSC stages.

As expected, we found a significant positive correlation between the classical subtype and TC+PKS and TC+KS mutants. This observation is important because SOX2 amplifications and alterations targeting the OSR pathway (*NFE2L2*, *KEAP1* and *CUL3*) occur far more frequently in this subtype (PMID:34358469).

Correlations between the classical subtype and mutants that maintain mucociliary differentiation (wild type, TC, TC+K, TC+PK, TC+P and TC+PS) are non-significant or negative.

Ideally, this correlation analysis should have been carried out with invasive and premalignant LUSC with genomic and transcriptomic profiling as, except for the TC+PKS mutant, our mutants mainly recapitulate those stages. Another limitation is the low tumour purity in the EMT-E and inflamed-secretory subtypes which hinders the analyses.

Nevertheless, we confidently conclude that mutants with concomitant SOX2 and KEAP1 mutations correlate with the classical subtype. A significant positive correlation was found with the EMT-E subtype, however further work is required to dissect the bias introduced by the low tumour purity in these samples. This could be achieved by microdissecting the tumour areas.

We are also showing the side-by side comparison requested by the reviewer in Supplementary Figure 8. We have compared the GO Biological terms regulated individually by each pathway in ALI cultures and LUSC samples with and without SOX2 amplifications, alterations in the OSR pathway and PTEN alterations. We identified significant overlap between the TC+S mutant and samples with SOX2 amplification and between the TC+K mutant and LUSC samples with somatic alterations in the OSR pathway. The comparison between the TC+P mutant and samples with somatic alterations was not significant. This lack of correlation is probably due to a much lower number of genes differentially expressed between samples with and without PTEN somatic alterations.

3. Establishing an in vitro model system could facilitate testing the cells for drug sensitivity. It would be relevant to show if this model systems could allow for drug screens and functional readouts. The authors could address how these different gene alterations may impact response to drugs (e.g. chemotherapy).

We thank the reviewer for this suggestion, as it will improve the scope of our model. To show that our model system allows for drug screens and functional readouts, we have treated wild-type and TC+PKS ALI cultures with three doses of cisplatin, a chemotherapy drug used to treat LUSC. We selected wild-type and TC+PKS ALI cultures as they recapitulate an optimal normal vs. tumour cell situation. In doing this, we expected a dose-dependent response to cisplatin and differences between the two genotypes. Using a multiplex assay to measure parameters correlated with cell viability and cell death, we detected dose-dependent changes in two cell death-related parameters, namely LDH activity in basal media and caspase 3/7 activity and a higher sensitivity to cisplatin in TC+PKS ALIs. The resazurin assay to measure viability did not perform well, possibly due to the high expression of reductases in TC+PKS mutants driven by the OSR pathway. Our results suggest that alternative methods to measure viability, possibly quantification of cellular ATP are more suitable. Therefore, we have demonstrated that our model system allows for functional readouts and can be used in drug screens. Moreover, our LUSC model is a more physiological system than submerged cultures as it recapitulates the intra-epithelial architecture and intercellular interactions that are likely to modify drug responses and include donor-matched controls.

Reviewer #3 (Remarks to the Author): Expert in lung squamous cell carcinoma clinical research and functional genomics

I am grateful to the Editorial team of Nature Communications for granting me and my junior colleague the opportunity to conduct a peer review of the manuscript authored by Ogden et al. The study successfully developed a new in-vitro human model for investigating the intrinsic phenotypes of genetically dysregulated pathways in LUSC. In accordance with the authors' expectations, I am of the opinion that this model has the potential to provide a robust means of understanding the complex relationships between genotype and phenotype. Furthermore, it can facilitate the development of target therapies and diagnostic strategies for LUSC.

We are grateful to the reviewer for their positive opinion about the manuscript and the comments made. We have now made several changes to the manuscript to address those comments. Those changes are discussed and specified below.

I would like to discuss several comments:

1. Many investigators have lately employed organoids for basic cancer research. It would be beneficial if the author could elucidate the pros and cons of this model in contrast to organoid in the discussion section.

We have now added a summary of the main advantages and disadvantages of our model in LUSC research in both basic and preclinical research. These changes can be found in the first paragraph of the discussion.

2. What is the authors' opinion on the applicability of this concept to cancer patients? For instance, can you induce normal cells to differentiate in a manner that replicates the specific characteristics of the cancer present in a specific patient?

The reviewer is introducing a very interesting discussion here. Overall, our concept can be applied for the benefit of cancer patients at multiple levels:

We have demonstrated that new therapeutic interventions can be discovered by building a new model. For instance, the strategy used in our manuscript is already pointing at potential therapeutic innovations that can be tested preclinically immediately, such as the targeting of NRF-2 inhibitors to patients with 3q amplification, or the use of those inhibitors to remodel the immune microenvironment and sensitising patients to immunotherapies.

We have demonstrated that our model allows the measurements of treatment readouts (developed in Supplementary Figure 9) and therefore, the use of the model as a preclinical platform in drug screens in monotherapies or combinations of therapies with chemotherapy.

The introduction of relevant somatic or germline variants associated with treatment resistance in the genetic manipulation strategy can assist pharmacogenomic preclinical studies to demonstrate the association and explore strategies to overcome that resistance.

The reviewer highlights the potential use of our concept to replicate the specific characteristics of the cancer present in a specific patient. Indeed, this was one of the motivations to develop our model: to recapitulate inter-patient heterogeneity observed in LUSC in the form of molecular and histological subtypes to identify vulnerabilities in those patients. In the next stage of our research, we will set out to model non-classical subtypes. Several strategies can be explored. For instance, club cells have been suggested to be the cell of origin of the inflamed subtype based on transcriptomic similarities. Our model is amenable to test this hypothesis as it would allow targeting of club cells using specific promoters to express Cas9. Additionally, certain somatic alterations appear to be more frequent in non-classical subtypes, namely, NF1 mutations. Manipulating hBECs to introduce those mutations and correlate the resulting phenotypes with non-classical subtypes constitutes another strategy to model other LUSC subtypes.

3. I concur with your viewpoint regarding the utilization of models for basic research. How do you foresee the implementation of this concept in clinical practice? What specific tests do you propose for acquiring hBECs from patients to build this model? If you successfully model cancer using hBECs from a particular patient, how do you envision its application in terms of diagnosis and treatment? Please engage in a discussion.

How do you foresee the implementation of this concept in clinical practice?

We have now demonstrated that we can use our model as a preclinical system to measure different cellular readouts of response to treatment with cytotoxic drugs. Therefore, the most immediate use of our model in clinical practice will be the screening of chemo- and radio-sensitising agents. However, our model will also allow the screening of new monotherapies, especially those causing cell death. Our model also enables us to use donor-matched wild-type cells as controls, as different sensitivities between donors might occur and would enable us to predict toxicities in normal epithelia.

Another interesting preclinical use of our model is the screening of drugs that regulate secreted immunomodulatory factors by analysing the basal media. Identifying drugs that modulate the secretion of immune modulatory factors might indicate a function in remodelling the immune microenvironment and sensitisation to immunotherapies. This approach can accelerate the pre-selection of these drugs for validation in-vivo.

Our new LUSC model is also an optimal system to identify cell surface biomarkers expressed in cancer cells and absent in normal epithelium. Cell surface biomarkers can serve multiple uses in LUSC early detection and therapy. They can be leveraged to direct treatments or diagnostic agents to cancer cells while sparing the normal epithelium. Namely, antibody-drug conjugates or CAR-T cells for LUSC treatment, and targeted radiolabelled or contrast agents (e.g. gold nanoparticles) to aid tumour imaging in LUSC early detection. Importantly, our system can facilitate the target-agnostic screening of cell-surface binders (e.g. selection of nanobodies using a phage display library) by selection of cancer cell binders with invasive or preinvasive mutants (TC+PKS, TC+KS or TC+S) and deselection of normal cell binders with wild type ALI cultures.

We also envisage that our model can be used in LUSC chemoprevention screens. Our results have shown that the expansion of clones with 3q amplification might be limited by activation of the OSR pathway. Using our model TC-PS model to test chemo preventive drugs and measuring the effect in the expansion of SOX2 clones constitutes an optimal experimental screen to test this type of drug.

What specific tests do you propose for acquiring hBECs from patients to build this model?

These tests will be dictated by the type of scientific questions to address. In general, primary hBECs should be tested for their capacity to differentiate into the proximal bronchial lineages- namely, club, goblet and ciliated cells- using organotypic air-liquid interphase cultures. However, one of the unique features of our concept is the modelling of predisposition and high-risk factors using hBECs from those populations. An example of this is COPD, an independent risk factor for LUSC. The bronchial epithelium of COPD patients exhibits alterations, such as goblet cell hyperplasia, that should be observable in organotypic cultures. In summary, the donor's history on respiratory diseases and carcinogen exposures (tobacco or occupational hazards) should be considered.

Additionally, testing for pre-existing genomic alterations is advisable, especially in hBECs derived from smoking donors due to the known accumulation of mutations in cancer drivers such as *TP53*. The need to implement this genomic characterisation also depends on the scientific questions, but it is especially advisable to exclude pre-existing alterations in the genes of interest. Additionally, pre-existing *TP53* mutations can hamper mutant selections with Nutlin-3a.

If you successfully model cancer using hBECs from a particular patient, how do you envision its application in terms of diagnosis and treatment? Please engage in a discussion.

This question can be answered from different angles.

The first angle is the modelling of LUSC from a patient using their own 'normal' hBECs. In this scenario, the 'individualised' model can be used to replace 'native' tumour samples in personalised medicine screens, should this material not be available due to a non-operable tumour or failure to expand the primary tumour ex-vivo (e.g. as organoids) or in vivo (as PDXs).

The second angle is the modelling of the generic LUSC subtypes. This is certainly the direction in which our model is going. Several molecular and histological subtypes have been described in

the literature. However, treatment is dictated by tumour stage and the presence of targetable mutations (e.g. EGFR mutations), and no evidence of the different sensitivities to existing therapies, or different vulnerabilities LUSC subtypes has been provided. As explained in the previous comment, we intend to use different approaches to model molecular subtypes with a prominent set of translational aims in mind.

With regards to diagnosis, we have previously discussed how to leverage our model to develop early detection strategies. However, these strategies might be specific for certain LUSC subtypes and fail in others. Therefore, developing models that recapitulate all or the majority of LUSC subtypes is essential to identify biomarkers for LUSC diagnosis and prioritise those with the potential to detect all LUSC subtypes.

Similarly, developing models that encompass LUSC subtypes will contribute to the improvement of precision LUSC medicine. Testing treatments in models that recapitulate different LUSC subtypes and identification of subtype-specific vulnerabilities. A plausible example of this principle is the function of Nrf-2 pathway activation in rescuing the detrimental interaction between SOX2 overexpression and PI3K/Akt pathway, which anticipates a higher effect of Nrf-2 inhibitors in patients with the classical LUSC subtype as this subtype exhibits high frequency of 3q-amplifications and mutations targeting the Nrf-2 pathway. Similar vulnerabilities will emerge during the development of non-classical subtype models.

4. The authors employed a genetic engineering technique, utilizing CRISPR-Cas9 for genetic modification in this model. How did the authors take measures to prevent the influence of off-target effects on their results? What measures were implemented to ensure the specificity of gene editing?

We apologise for not adding this to the Materials & Methods section. To design the sgRNAs for this project, we used the Benchling tool (Benchling.com) which provides sgRNA designs with on-target and off-target scores. We selected batches of four sgRNAs with the lowest off-target score for each gene, prioritising sgRNAs targeting the ORF upstream the first annotated functional domain or within it to ensure complete inactivation of the protein. To discard targeting other loci, we took an approach akin to Drost and colleagues (PMID:25924068). The top five predicted off-target loci with the highest off-target scores using Sanger sequencing. With the exception already mentioned in the manuscript, indels were not observed in any other predicted off-target sites. Additionally, we have analysed the levels of PTENP1 lncRNA to assess the potential effect of the PTENP1 indels on RNA expression and/or stability. Our qPCR analysis showed very low levels PTENP1 that were not affected by the presence of PTENP1 indel (TC+P mutant).

5. The study discovered that overexpression of SOX2 inhibits proliferation of cells in 2D cultures, but enhanced anchorage-independent growth and invasiveness. How do the authors reconcile this paradox, and what further experiments might clarify the role of SOX2?

In our opinion, the inhibition of proliferation is not necessarily a paradox. Indeed, SOX2 slows but does not abrogate proliferation. This anti-proliferative effect of SOX2 is not new and has already been observed in other systems (PMID: 28388544). SOX2 regulates the expression of thousands of genes and is likely to induce transformation by several mechanisms simultaneously.

As suggested by the reviewer, further experiments can be carried out to deconvolve the mechanisms that are relevant for transformation. This endeavour requires a separate and systematic approach that constitutes an extensive project. Genetic screens to target SOX2-regulated genes in LUSC models with and without SOX2 amplification can be used to address the

key regulators of SOX2-driven tumourigenesis. However, SOX2 is likely to induce tumourigenesis by cell-autonomous and TME mediated effects and identifying all these mechanisms might be difficult. Another approach is to identify the genes that are directly regulated by SOX2 using CHIP-seq. This will result in a smaller list of targets and more accurate identification of biological processes regulated by SOX2. This could be used to inform a hypothesis-driven set of experiments to manipulate SOX2-regulated processes.

However, we can hypothesise what mechanisms are likely to be relevant in SOX2-driven LUSC. SOX2 primarily drives a drastic change in cell and epithelial identity, from a typical pseudostratified bronchial morphology to a stratified squamous morphology. This change enables the transformed epithelium to lose its monolayer (pseudostratified) structure and acquire a stratified three-dimensional organisation more akin to a tumour situation that enables unrestricted cell expansion without contact inhibition. We estimate that the number of cells in TC+PKS ALI cultures increases 4-fold compared to wild type.

Additionally, we have confirmed changes in antigen presentation molecules (MHC-II), confirming that SOX2 directly or through squamous differentiation could mediate the transition to a more immunosuppressive environment. Reports by previous authors confirm this function.

Also, SOX2 likely induces the expression of genes whose products prime cells for invasion when other alterations arise (e.g. activation of PI3K/Akt pathway). Indeed, our RNA-seq analysis has shown up regulation of GPNMB by SOX2, a protein previously involved in cell migration and invasion (PMID:28295306, PMID:2663634). Additionally, multiple authors have reported an anti-correlation between invasion and proliferation that also might explain this apparent paradox of SOX2 function (PMID: 27634432) as an oncogene.

An interesting hypothesis is that cells with SOX2-amplification outcompete normal cells in the bronchial epithelium by inhibiting the growth and/or fitness of normal epithelial cells, resulting expansion of premalignant cells at the expense of normal cells without changes in proliferation. Our system provides a good opportunity to address this using organotypic air-liquid interface cultures.

In summary, SOX2-amplification in LUSC is likely to act through multiple pro-oncogenic alterations including the changes in tissue architecture, invasion/migration, cell competition and immune evasion previously mentioned. Downregulation of proliferation might result in longer tumour latency; however, the balance is likely favourable towards tumour expansion.

Although a shorter latency time in SOX2 driven tumours has not been reported, well-differentiated (keratinising) LUSC exhibits a better prognosis (PMID:25189482) and the classical subtype is enriched in this histological subtype (35358469). This association suggests a link with SOX2.

6. The results demonstrate heterogeneity among donors, especially in terms of colony formation and the expansion of p63+ cells. How do the authors interpret and clarify the variation between donors, and what are the implications for the generalizability of the findings?

We have indeed considered different explanations for the heterogeneous data that we have observed in some of the assays. Donor 3 was arguably the most discordant of the three donors. This discordance manifested mainly in high levels of anchorage-independent growth and p63+cell expansion in the TC+KS mutant, similar to those observed in the TC-PKS mutant. Although this might suggest that the transformation of donor 3 hBECs might be independent of

PI3K/Akt pathway activation, the collagen-disc invasion assay, an assay that directly assesses invasiveness, did show that donor 3 TC+PKS mutants were significantly more invasive than TC+KS, confirming that activation of PI3K-Akt pathway is indeed required for complete transformation of donor 3. Although explaining the higher-than-expected levels of anchorage-independent growth and p63+ cell expansion would require detailed experimental analysis, it might suggest inter-patient heterogeneity in the rate of development and evolution of premalignant lesions due to patient-intrinsic factors such as germline and/or somatic mutations, specific environmental exposures, ethnicity and sex (donor 3 is the only female of the three donors). Should this be the case, donor 3 is predisposed to have faster-progressing premalignant lesions. However, technical explanations are also possible, such as differences in the bronchial regions where the hBECs were sampled.

Notwithstanding the former patient-intrinsic and technical differences, we observed that a minority of classical LUSC patients from the CPTAC cohort (4/24, 16.6%) do not harbour somatic alterations in PI3K/Akt components. This observation might be explained by somatic activation of the pathway by epigenetic mechanisms or that a minority of patients develop LUSC independently of PI3K/Akt pathway activation, possibly due to germline factors. The later explanation would be partly supported by our results, and a larger cohort is necessary to address this hypothesis.

7. The study focuses mainly on the SD, PI3K/Akt, and OSR pathways. Are there any further pathways or genetic alterations that could have a significant impact on LUSC but were not taken into account in this model? Do authors have precise criteria for selecting three pathways for the creation of this model?

As the reviewer correctly states, we aimed to model the most frequently dysregulated processes and pathways rather than individual alterations. When we set out to design our strategy, the TCGA LUSC dataset (PMID:22960745) was the most detailed and systematic account of the LUSC genomic landscapes and reported the SD, PI3K/Akt, and OSR pathways as the most frequently targeted by mutually exclusive somatic alterations and later reports have not contradicted this observation. Therefore, frequency was the main criterion for selecting the three pathways. Including only one more mutation in the experimental design would have resulted in 17 mutants per donor, a number that would have doubled the time and resources required for the project.

However, as correctly suggested by the reviewer, other significantly mutated genes that are not components of those three pathways are frequent and worth being investigated using our model. A clear example are mutations in the epigenetic regulators *KMT2D* and *ARID1A*. The function of this *KMT2D* in LUSC has already been investigated in the magnificent study by Pan and colleagues (PMID:36525973), although, as stated in our response to reviewer #2, there are multiple outstanding questions about the mechanisms whereby this chromatin modifier drives LUSC and the LUSC subtypes it drives.

We also think that the function of alterations targeting the RTK-RAS pathway should be investigated to discern whether they drive LUSC, as these alterations (mainly *NF1*, *KRAS* and *EGFR* mutations) also occur in LUSC, although to a lesser frequency than SD, PI3K/Akt, and OSR. Specifically, the CPTAC consortium data has shown that *NF1* mutations occur preferentially in non-classical LUSC subtypes, which might indicate a role in driving those subtypes. On the other hand, unpublished data reported at the AACR meeting 2024, have shown frequent mutations RTK-RAS pathway components in normal airway epithelium and a lower frequency than expected of some of these mutations in lung cancer. This indicates a 'protective' function for those

mutations similar to *NOTCH1* inactivations in the epidermis. Our system will be instrumental in addressing the latter points when that data is finally published.

8. LUSC typically originates from cells that have been exposed to chronic inflammation, such as in heavy smokers or individuals with chronic obstructive pulmonary disease (COPD). I am curious about the potential alterations in the outcomes if this model were to be implemented on hBEC derived from those specific patients. What are the authors' opinions regarding this matter? Is there any likelihood that the LUSC model derived from heavy smoker donors exhibits similarity to the LUSC of actual patients?

This is a crucial point that we have repeatedly considered in our modelling concept. One of the motivations behind our model was to unpick the mechanisms of high risk and predisposition by manipulating hBECs derived from those populations to constitute a more accurate approach.

Our model is especially suitable to investigate the function of epigenetic changes driven by cigarette smoke on LUSC initiation and progression.

Multiple articles have reported the effect of cigarette smoke on epigenomes and used exposure to cigarette smoke condensate as an experimental model to address the multiple mechanisms whereby smoking fuels cancer. However, lung cancer patients have smoked for years or even decades before the onset of cancer, and recapitulating this long-term exposure is not experimentally feasible. On the other hand, isolating hBECs from long-term smokers might overcome that limitation. Direct comparison of phenotypes in genetically engineered hBECs isolated from non-smokers and long-term smokers is arguably the best experimental system to unravel the function of non-genetic effects in tumorigenesis. However, the normal epithelium of smokers harbours mutations in known cancer genes such as *TP53*, and isolating hBECs from a sufficient number of smoking donors without mutations in cancer genes to segregate the effect of non-genetic alterations from that of pre-existing genetic alterations might be difficult.

Similarly, COPD is known to be an independent risk factor of lung cancer, including LUSC and isolating hBEC from patients with and without COPD to investigate how this condition affects genotypes could shed light on the mechanisms that determine such risk. However, segregating the effect of smoking from that of COPD might be difficult as most COPD patients are also smokers. This problem could be overcome by comparing the phenotypes of genetically engineered hBECs from never-smokers, long-term smokers and smokers with COPD.

9. The authors stated that the amplification of *SOX2* is linked to the remodeling of the tumor microenvironment. Nevertheless, this particular model lacks TME, and the authors drew this conclusion based on the downregulation of *CIITA*. I fully disagree with this interpretation.

The reviewer is completely right in highlighting that our remarks in this regard are overstated. We have changed the text of the discussion to highlight that *SOX2* regulates genes involved in regulators of immunity (*CIITA*) and other factors with TME remodelling (serin-protease inhibitors) function and that experimental confirmation of that role is necessary due to its likely relevance in cancer therapy.

Changes are as follows:

- **In the following sentence, we have introduced the changes in bold:** *Our analysis of SOX2-associated expression data strongly indicates that SOX2 amplification in LUSC is multi-faceted and constitutes a hub of pro-tumourigenic cues, not limited to cell*

autonomous processes (squamous differentiation, RTK-RAS pathway activation), but also likely to remodel the immune microenvironment.

- **The sentence** “*For the first time, we have provided evidence that places SOX2 at the apex of cell intrinsic and TME-mediated pro-oncogenic signals that include links with innate and acquired immunity in LUSC such as MHC-II downregulation and neutrophil-elastase inhibitors*” **has been replaced with** “*Of note, regardless of the focus of our results on cell-autonomous effects of SOX2, we have provided evidence that SOX2 regulates gene expression programs that are likely to regulate innate and acquired immunity in LUSC such as MHC-II downregulation and neutrophil-elastase inhibitors. Further research is needed to assess the functional consequences of those changes in immunosurveillance and tumour progression.*”

10. I believe that the results of WGCNA are not the main findings of this study. What if the authors relocate the results as additional or supplementary data?

We concur with the reviewer that the WGCNA results are not the most important findings of the study. However, for transparency and quality control purposes, we would like to keep it in the main text of the manuscript and the data will be part of the supplementary information.

POINT-TO-POINT RESPONSE TO THE REVIEWER'S COMMENTS (NCOMMS-24-11773B)

REVIEWERS' COMMENTS

Reviewer #1 (Remarks to the Author):

The revisions have substantially improved the manuscript and addressed several of the key concerns raised by this reviewer. The inclusion of additional data validating the human relevance of the model using analyses of TCGA and CPTAC datasets has strengthened the validity of the developed model. Although some limitations remain, especially those related to capturing heterogeneity and the impact of the tumor microenvironment, overall, the revisions have enhanced the clarity and quality of the paper in providing an additional model system to study LUSC. Therefore, I recommend publication in Nature Communications.

We thank you Reviewer #1 for the previous and current comments about the increase in clarity and quality in the revised manuscript. The comments from the reviewer were crucial to achieve these improvements. Likewise, we acknowledge the limitations regarding heterogeneity and tumour microenvironment. We are now using this LUSC-modelling strategy to interrogate the role of stromal cells and to model non-classical LUSC subtypes.

Reviewer #2 (Remarks to the Author):

Ogden et al. revised their manuscript addressing most of the comments suggested by the reviewers. Overall, the authors provide a good amount of additional data. I have no further comments.

We are very grateful to Reviewer #2 for the comments to the first submission which helped us improve the manuscript.

Reviewer #3 (Remarks to the Author):

I carefully reviewed the updated version and determined that all the weaknesses I stated were properly addressed. While the model's applicability in clinical settings remains uncertain, it appears to hold potential for basic research.

We thank Reviewers #3 and #4 for the comments to the first submission and the revised manuscript, which contributed to improve our work. Whereas we agree that the clinical applicability of the model is uncertain, the potential of our LUSC-modelling strategy in basic and preclinical research is where the maximal value of the model lies. We acknowledge the value of co-review for Early Career Researchers training and recognise the great insight of Reviewer #3 into the topic.

Reviewer #4 (Remarks to the Author):
